# Physical and biogeochemical impacts of RCP8.5 scenario in the Peru upwelling system

Vincent Echevin[1], Manon Gévaudan[1,2,3], Dante Espinoza-Morriberón[2], Jorge Tam[2], Olivier Aumont[1],
Dimitri Gutierrez[2,4], François Colas[1,2]

[1]Sorbonne Université, LOCEAN-IPSL, CNRS/IRD/MNHN, 4 place Jussieu, 75252 Paris, France

[2] Instituto del Mar del Peru (IMARPE), Esquina General Gamarra y Valle, Callao, Perú

[3]Laboratoire d'Etudes en Géophysique et Océanographie Spatiales, 14 av. E.Belin, 31400 Toulouse, France

[4]Laboratorio de Ciencias del Mar, Universidad Peruana Cayetano Heredia, Lima, Perú

*Correspondence to*: Vincent Echevin (vincent.echevin@ird.fr)

**Abstract:**

The northern Humboldt current system (NHCS or Peru upwelling system) sustains the world's largest small pelagic fishery. While a nearshore surface cooling has been observed off southern Peru in recent decades, there is still considerable debate on the impact of climate change on the regional ecosystem. This calls for more accurate regional climate projections of the 21[st] century, using adapted tools such as regional eddy-resolving coupled biophysical models. In this study three coarse-grid Earth System Models (ESMs) from the Coupled Model Intercomparison Project (CMIP5) are selected based on their 20   biogeochemical biases upstream of the NHCS and simulations for the RCP8.5 climate scenario are dynamically downscaled at ~12 km resolution in the NHCS. The impact of regional climate change on temperature, coastal upwelling, nutrient content, deoxygenation and the planktonic ecosystem is documented. We find that the downscaling approach allows to correct major physical and biogeochemical biases of the ESMs. All regional simulations display a surface warming regardless of the coastal upwelling trends. Contrasted evolutions of the NHCS oxygen minimum zone and enhanced stratification of phytoplankton are found in the coastal region. Whereas trends of downscaled physical parameters are consistent with ESM trends, 25   downscaled biogeochemical trends differ markedly. These results suggest that more realism of the ESMs circulation, nutrient and dissolved oxygen fields is needed in the eastern equatorial Pacific to gain robustness in the projection of regional trends in the NHCS.

**1 Introduction**

Eastern Boundary Upwelling Systems (EBUS) are oceanic systems where alongshore winds generate the upwelling of deep, cold and nutrient-replete waters. This drives a high  biological productivity and thriving small pelagic fisheries which are major sources of income for the adjacent countries. In particular, the Peruvian Upwelling System (also known as the

Northern Humboldt Current System, NHCS in the following), located in the South Eastern Tropical Pacific, is the most productive EBUS in terms of fish catch (Chavez et al., 2008), due to its rich anchovy fishery. Moreover, the subsurface water masses in the NHCS are located in the poorly ventilated so-called "shadow zone" of the south eastern Pacific (Luyten et al., 1983). This low ventilation creates a subsurface water body with very low oxygen concentration, the oxygen minimum zone (OMZ). The OMZ results from a balance between oxygen consumption by respiration of large amounts of organic matter exported from the highly productive surface layer, and ventilation by the equatorial current system composed of eastward jets transporting relatively oxygenated waters (Czeschel et al., 2011; Montes et al., 2014). A particular aspect of the NHCS OMZ is its very low oxygen concentration (anoxia) at relatively shallow depths, which impacts the local marine ecosystem (Stramma et al., 2010; Bertrand et al., 2011).

In the recent decades, public concern has risen about the impact of climate change on EBUS. Using ship wind observations, Bakun (1990) showed that upwelling-favorable winds increased over recent decades (1950-1990) in several EBUS. He proposed that nearshore winds would continue to intensify due to an enhanced differential heating between land and sea, driven by a stronger greenhouse effect over land. However, this hypothesis has been challenged in the NHCS because of observation bias (*e.g.* Tokinaga and Xie, 2011) and poleward displacement of the South Pacific Anticyclone (Belmadani et al., 2013; Rykaczewski et al ., 2015). Nevertheless, in situ and satellite Sea Surface Temperatures (SST) show a conspicuous surface coastal cooling off southern Peru (15°S) since the 1950s. This cooling, consistent with a wind increase found in ERA40 reanalysis, suggests a possible intensification of the wind-driven upwelling (Gutierrez et al., 2011).

Recent analysis of the 5[th] Coupled Model Intercomparison Protocol (CMIP5)  global circulation models (GCMs) reported that the intensification of nearshore winds under scenarios of carbone dioxyde concentration increase is mainly confined to the poleward portions of EBUS (Wang et al., 2015; Rykaczewski et al., 2015, Oyarzún and Brierley, 2019). However, the evolution of winds in the NHCS remains unclear (note that the NHCS stricto sensu was not included in these studies). Furthermore, the realism of IPCC GCMs is hampered by the coarse resolution of the model grids (~100-200 km), that does not allow to represent the details of coastal orography and coastline that influence the coastal wind structure.

A few downscaling studies focusing on regional wind changes in the NHCS have provided invaluable information. NHCS upwelling-favorable winds may weaken in the future, mainly during the productive austral summer season (Goubanova et al., 2011; Belmadani et al., 2014). However, only idealized extreme scenarios (preindustrial, doubling (2xCO2) and quadrupling (4xCO2) of carbon dioxyde concentration) from a single GCM (IPSL-CM4, Marti et al., 2010) of the Intergovernmental Panel on Climate Change (IPCC) Fourth Assessment Report (AR4) were downscaled in these studies. In line with these studies, Echevin et al. (2012) used a regional ocean circulation model (RCM) forced by statistically downscaled atmospheric winds from Goubanova et al. (2011) to downscale the NHCS ocean temperature and circulation changes under 2xCO2 and 4xCO2 scenarios. They found a strong warming in the surface layer, of up to ~+5°C nearshore in the 4xCO2 scenario with respect to preindustrial conditions, and an upwelling decrease during austral summer. Following the same regional modeling approach and using the downscaled winds from Belmadani et al. (2014), Oerder et al. (2015) found a year-round reduction in upwelling intensity, mitigated by an onshore geostrophic flow. The shoaling of upwelling source waters in the 2 scenarios suggests that upwelled waters could become less nutrient-rich, and thereby reduce nearshore primary productivity (Brochier et al., 2013).

The impact of climate change on the NHCS productivity, oxygenation and acidification has been even less investigated. Assuming Bakun's (1990) hypothesis of increasing coastal winds, Mogollón and Calil (2018) found a moderate increase (5%) of the NHCS productivity using a RCM. However, they did not take into account the large-scale stratification changes driven by climate change that may significantly contribute to nearshore stratification and mitigate the upwelling (Echevin et al., 2012; Oerder et al., 2015). Following a similar approach, Franco et al. (2018) found a sustained acidification of NHCS shelf and slope waters under the Representative Concentration Pathway 8.5 scenario (RCP8.5, the so-called worst case AR5 climate scenario corresponding to a 8.5 Wm$^{-2}$ heat flux driven by the greenhouse effect, *e.g.* van Vuuren et al., 2011), driven by changes in surface fluxes of atmospheric $CO_2$ concentration and subsurface dissolved inorganic carbon concentrations. However, as in Mogollón and Calil (2018), the impact of climate change on NHCS surface winds, circulation and stratification was unaccounted for in Franco et al. (2018).

In brief, previous regional modelling experiments were either obtained from (i) the downscaling of one single GCM or Earth System Model (a GCM including a biogeochemical model, hereafter ESM), (ii) the analysis of relatively short time periods (e.g 30 years in the stabilized phase of the 2xCO2 and 4xCO2 scenarios in Echevin et al., 2012; Oerder et al., 2015; Brochier et al., 2013), or (iii) simplified approaches that did not account for all physical forcings (*e.g.* Mogollón and Calil, 2018; Franco et al., 2018). More work is thus needed to evaluate the robustness of these findings under climate scenarios taking into account economic and population growth assumptions (*e.g.* RCP8.5) and over longer time periods (*e.g.* 100 years).

In the present work, 3 different ESMs are dynamically downscaled in the NHCS using a regional coupled dynamical-biogeochemical model. The studied time period is 2005-2100 under the RCP8.5 scenario. The regional trends from RCMs are compared to illustrate the diversity of climate change regional impacts. RCM trends are also contrasted with those of the ESMs in order to highlight the impact of the downscaling process. In the next section (section 2) the regional model, the selection process of ESMs and the downscaling methodology are described. Results are presented in section 3: we describe the trends of key physical and biogeochemical parameters such as temperature, coastal upwelling, thermocline depth, oxygenation, nitrate and productivity. The approach and implications of our work are discussed in section 4. The conclusions and perspectives are drawn in section 5.

## 2 Methodology

### 2.1 Ocean model

The Regional Ocean Modeling System (ROMS) was used to simulate the ocean dynamics. The ROMS AGRIF (version v3.1.1 is used in this study) resolves the Primitive Equations, which are based on the Boussinesq approximation and hydrostatic vertical momentum balance (Penven et al., 2006; Shchepetkin and McWilliams, 2009). A fourth-order centered advection scheme allows the generation of steep tracer and velocity gradients (Shchepetkin and McWilliams, 1998). For a complete description of the model numerical schemes, the reader can refer to Shchepetkin and McWilliams (2005).

The model domain spans over the coasts of south Ecuador and Peru from 5°N to 22°S and from 95°W to 69°W. It is close to the one used in Penven et al. (2005). The horizontal resolution is 1/9°, corresponding to ∼12 km. The bottom topography from STRM30 (Becker et al., 2009) is interpolated on the grid and smoothed in order to reduce potential errors in the

horizontal pressure gradient. The vertical grid has 32 sigma levels.

Wind speed, air temperature and humidity, and ROMS SST are used to compute latent and sensible heat flux online using a bulk parameterization (Liu et al., 1979).

**2.2 Biogeochemical model**

ROMS is coupled to the Pelagic Interaction Scheme for Carbon and Ecosystem Studies (PISCES) biogeochemical model. PISCES simulates the marine biological productivity and the biogeochemical cycles of carbon and main nutrients (P, N, Si, Fe; Aumont et al., 2015) as well as dissolved oxygen (DO) (*e.g.*, Resplandy et al., 2012, Espinoza-Morriberón et al., 2019). It has three non-living compartments which are the semi-labile dissolved organic matter, small sinking particles and large sinking particles, and four living compartments represented by two size classes of phytoplankton (nanophytoplankton and diatoms) and two size classes of zooplankton (microzooplankton and mesozooplankton). The ROMS-PISCES coupled model has been used to study the climatological (Echevin et al., 2008), intraseasonal (Echevin et al., 2014), and interannual variability of the surface productivity (Espinoza-Morriberón et al., 2017) and oxygenation (Espinoza-Morriberón et al., 2019) in the NHCS. Detailed parameterizations of PISCES (version 2) are reported in Aumont et al. (2015). Note that we used an earlier version of the model (PISCESv0) in this study, as PISCESv2 had not been coupled to ROMS yet at the beginning of our study. Here we describe the following parameterizations of PISCESv0: i) diatoms and nanophytoplankton growth, microzooplankton grazing and mortality, mesozooplankton mortality depend on temperature (T) and are proportional to $e^{a.T}$ with a=0.064 °C$^{-1}$; ii) mesozooplankton grazing on nanophytoplankton and diatoms is proportional to $e^{b.T}$ with b=0.076 °C$^{-1}$. These differences, in particular the larger temperature-enhanced mesozooplankton grazing with respect to phytoplankton growth, can play an important role in the context of surface warming in the NHCS. Boyd et al. (1981) measured grazing of Peruvian copepods, however further laboratory experiments are needed at different temperatures to calibrate these rates.

**2.3 Selection of the Earth System Models**

Three CMIP5 ESMs are selected for the regional downscaling. The selection process is based on the nutrients simulated by the ESMs and on the evaluation of biogeochemical bias. Only five ESMs (CNRM, GFDL, IPSL, CESM and Nor-ESM) represent the four nutrients (silicate, phosphate, nitrate and iron) and DO required by PISCES. As different ESM versions were available, a total of eight ESMs (CNRM-CM5, GFDL-ESM2M, GFDL-ESM2G, IPSL-CM5A-MR, IPSL-CM5A-LR, IPSL-CM5B-LR, CESM1, Nor-ESM1-ME) were compared to observations from the World Ocean Atlas (WOA2009, Fig.1). Following Cabré et al. (2015), the ESM DO, nutrients, temperature and salinity were averaged at 100°W between 5°N and 10°S, near the location of the western open boundary of the RCM, for the period 1980-2005 (1950-2005 for T and S). This meridional section intersects eastward jets: the Equatorial Undercurrent (EUC) at 0°S and the off-equatorial Southern Subsurface Countercurrents (SSCCs) at ~ 4°S and ~ 8°S (Montes et al., 2010). These jets transport physical and biogeochemical properties to the Peru upwelling region (Montes et al., 2010, 2014; Oerder et al., 2015; Espinoza-Morriberón et al., 2017, 2019).

Visual examination of the ESMs temperature and salinity profiles (Fig. 1) suggests that the corresponding biases are weak in comparison with other variables. The comparison between the biases of different variables can be quantified by

computing a bias normalized by the mean state, averaged between 0 and 500 m depth (0-250m for temperature and salinity, see Table 1). The ESMs normalized temperature bias is weaker than the biogeochemical biases (Table 1).

All ESMs simulate an oxygen decrease with depth (Fig.1a), but oxygen values are too low (i.e. <10 μmol l$^{-1}$) in CESM1-BGC, GFDL-ESM2M, GFDL-ESM2G and NorESM1-ME. Slightly negative values are attained below 300 m depth for GFDL-ESM2G. CNRM-CM5. In contrast, the 3 IPSL model versions, which all include PISCES as biogeochemical component, overestimate the oxygen content above ~600 m depth. Note that only CESM1-BGC is able to reproduce the observed oxygen increase below 400 m depth, which corresponds to the lower limit of the OMZ.

In terms of nitrate concentration, the most realistic models in the upper 300 meters are GFDL-ESM2G, GFDL-ESM2M and CESM1-BGC (Fig.1b). However, the model biases become negative and increase strongly at depths greater than 300 m. A negative bias found in the three IPSL ESMs (~ 3-4 μmol l$^{-1}$ for IPSL-CM5A-MR and IPSL-CM5A-LR and ~6-8 μmol l$^{-1}$ for IPSL-CM5B-LR) is roughly constant over depth. CESM1-BGC, GFDL-ESM2G and NorESM1-ME display too low nitrate concentrations below 250 m depth, possibly due to denitrification in the anoxic OMZ (Fig.1a).

The GFDL-ESM2M phosphate profile is very close to the observations (Fig.1c, Table 1), whereas the three IPSL ESMs and CNRM-CM5 underestimate phosphate concentrations with a roughly constant bias over depth (negative bias of ~0.5-1 μmol l$^{-1}$). In contrast, NorESM1-ME, GFDL-ESM2G and CESM1-BGC overestimate the phosphate concentrations.

The IPSL and CESM1-BGC silicate profiles are close to observations above ~250 m depth, whereas the positive bias in GFDL-ESM2M and NorESM1-ME increases below 200 m depth. The CNRM-CM5 negative bias is moderate between 50 and 300 m depth (Fig.1d).

To conclude, as the three IPSL ESMs and the CNRM-CM5 include the PISCES biogeochemical model also used in the regional simulations and provide reasonable nutrient bias with respect to the other ESMs (Table 1), IPSL-CM5A-MR (which nitrate and phosphate bias is weaker than the two others IPSL ESMs, Table 1) and CNRM-CM5 are selected. We also select GFDL-ESM2M, which represents well the nitrate and phosphate profiles in the upper layers, and whose bias did not increase at depth as in GFDL-EMS2G. CESM1-BGC also has weak biases with respect to the latter ESMs (Table 1), but some variables were not available from the archive (*e.g.* 10 m wind) at the beginning of this study. We thus restrict our study to the downscaling of three ESMs. The main characteristics of the selected ESM ocean models (grid spacing and biogeochemical structure) are summarized in Table 2. We refer to the ESMs as CNRM, IPSL and GFDL in the following sections and figures.

### 2.4 Atmospheric forcing methodology:

A bias correction is used to construct monthly forcing files (*e.g.* Oerder et al., 2015; note that daily files were not available for all ESMs). For each forcing variable X (i.e. X=wind velocity, air temperature, ...), the bias-corrected variable X' is computed as follows:

$$X' = X_{OBSclim} + (X_{ESM-RCP8.5} - X_{ESM-hist-clim}) \qquad (1)$$

$X_{OBSclim}$ corresponds to a monthly climatology of observed values, $X_{ESM-RCP8.5}$ corresponds to the coarse-grid ESM values for each month, and $X_{ESM-hist-clim}$ to a monthly climatology of the coarse-grid ESM values during the historical period (2000-2010). This allows substracting the ESM mean bias, assuming that it remains identical over the historical period and over 2000-2100. This method has been used in several papers (Cambon et al. 2013; Echevin et al. 2012; Oerder et al., 2015). The SCOW

(Risien and Chelton, 2008) surface wind and COADS (Da Silva et al., 1994) downward shortwave and longwave fluxes and air parameters (temperature and specific humidity) climatologies were used for $X_{OBSclim}$. Note that submonthly wind variability may impact significantly surface chlorophyll in other EBUS, such as off northern California where the wind variability is much stronger than off Peru (*e.g.* Gruber et al., 2006). Indeed, a previous regional modelling study in the NHCS showed a weak impact (less than 10% difference) of daily wind stress with respect to monthly wind stress on 7-year-averaged biogeochemical variables (Echevin et al., 2014). This suggests that using monthly winds may not  impact significantly the climate trends reported in this study.

**2.5 Open boundary and initial conditions for physics and for biogeochemistry**

As in Echevin et al. (2012) and Oerder et al. (2015), the ESM monthly sea level, temperature, salinity, horizontal velocity at the locations of the RCM open boundaries are directly interpolated on the model grid without bias correction. Given the important bias of the ESM mean biogeochemical state (*e.g.* Bopp et al., 2013; Cabré et al., 2015), we apply the bias correction described in Eq.(1): we add the WOA2009 (1°x1°) monthly climatology of the biogeochemical variables (nitrate, silicate, phosphate, iron, DIC, DOC, alkalinity, oxygen) and the annual mean anomalies (see Eq.(1)). The 3D fields were interpolated on the ROMS grid using the ROMSTOOLS package (Penven et al., 2008).

The three simulations are initialized as follows. Initial conditions from the ESM physical parameters of the historical simulation (2000-2010 January average) and WOA biogeochemical values (January) constitute the initial state.  A 9-year spin up simulation from 1997 to 2005 is then performed to reach equilibrium. The runs are then forced by RCP8.5 conditions until 2100. State variables and biogeochemical rates (*e.g.* primary production) are stored every 5 days. The regional simulations are named R-IPSL, R-CNRM, R-GFDL in the following.

**2.6 Additional data sets**

Two ocean reanalysis products are used to evaluate the ESM equatorial circulation and thermocline in present conditions. The SODA 2.3.4 reanalysis (Carton and Giese, 2008) over the period 1992–2000 assimilates observational data in a general circulation model with an average horizontal resolution of 0.25°. The recently available GLORYS12V1 reanalysis over the period 1993-2017 is also used (Ferry et al., 2012). Altimeter data, in situ temperature and salinity vertical profiles and satellite SST were jointly assimilated in GLORYS12V1 (Lellouche et al., 2018). This product is freely distributed by the Copernicus Marine Environment Monitoring Service.

Several sets of observations are used to evaluate the realism of ESMs and RCMs in present conditions (i.e. 2006-2015 period). In situ data include CARS2009 gridded fields of temperature, nitrate and oxygen (0.5°x0.5°, Rigway et al., 2002), high resolution (0.1°x0.1°) regional monthly climatologies of temperature (Grados et al., 2018) and oxygen (Graco et al., 2018) including measurements collected during IMARPE (The Sea Institute of Peru) cruises. AVHRR satellite SST (2006-2015) is  used to assess the RCM SST. Surface chlorophyll-a monthly climatologies from SeaWIFs (1997-2010) and MODIS (2002-2015) are used to evaluate the RCM surface chlorophyll.

**2.7 Coastal indices**

Time series of coastal indices characterizing the variability over the central Peru shelf for specific variables are computed. The variables are averaged in a coastal band extending from the coastline to 100 km offshore and between 7°S and 13°S.

An index of coastal upwelling, the cross-shore transport in a coastal band, is computed from the model output (Colas et al., 2008; Oerder et al., 2015; Jacox et al., 2018). The mean horizontal transport is computed each month in a coastal strip extending from 7°S to 13°S and from the coast to 100 km offshore. The transport is integrated vertically over the Ekman layer depth. The latter is diagnosed as follows: we compute the surface geostrophic current using model sea surface height, and integrate the thermal wind relationship from the surface to the depth (equal to Ekman layer depth) at which the cross-shore current and the cross-shore geostrophic current differ by less than 10% (see Oerder et al., 2015 for more details). The computation of this index is more straightforward than one based on model vertical velocity (Jacox et al., 2018) and leads to similar values (*e.g.* see Fig.4 in Jacox et al., 2018). In contrast with coastal upwelling indices based on Ekman transport only, this index takes into account the role of the cross-shore geostrophic current which can modulate the coastal upwelling (*e.g.* during El Niño events, Colas et al., 2008; Espinoza-Morriberón et al., 2017).

### 2.8 Statistical methods

Only time scales longer than 5-7 years (*e.g.* El Niño time scales) are studied in this work. Therefore the time series are low-pass filtered using a ten-year moving average. This allows to filter the ENSO variability which is very strong in the NHCS, but not the focus of the present study. Linear trends of the time series are computed using a least squares method. The percentage of change between 2006 and 2100 associated with the linear trends are listed in 3 Tables (Table 3 for physical variables, Table 4 for oxygen and nitrate, and Table 5 for chlorophyll and zooplankton). Statistical significance is presented as a 90% confidence interval, based on a bootstrap method: we compute a 10 000 member synthetic distribution derived by randomly removing data in the annual series. The confidence limits of the trends are converted into confidence limits for the percentages reported in the tables.

### 3 Results

In the following sections we show that the RCM is able to represent the main characteristics of the NHCS coastal upwelling system thanks to its high spatial resolution (relatively to the ESMs) and to the bias correction of the forcing. We then describe the long-term trends over the period 2006-2100 under the RCP8.5 scenario for key downscaled physical (surface and subsurface temperature, heat and momentum fluxes, upwelling) and biogeochemical parameters (oxygen and nutrient content, primary productivity, planktonic biomass) in the upwelling system but also in the equatorial band offshore of the NHCS. For selected variables we also compare the downscaled simulations and the coarse-grid ESMs. In the next section, we first characterize the downscaled physical fields.

### 3.1 Physical mean state and variability
**Sea Surface Temperature spatial patterns**

We first contrast the Sea Surface Temperature (SST) patterns of the ESMs and RCMs to highlight the efficiency of the dynamical downscaling. The actual observed SST displays the cold water tongue along the coast and associated cross-shore SST gradient, characteristic of coastal upwelling (Fig.2a). The RCM simulates correctly these upwelling features (Fig.2b). The fine representation of the coastline, shelf and slope topography and bias-corrected alongshore winds (see section 2.4) all play a role in the correct representation of the upwelling structure. The upwelling vertical structure is also well reproduced in
the RCMs. Mean cross-shore temperature profiles (within 500 km from the coast and between 7°S and 13°S) display the typical nearshore isotherms shoaling in the 0-100m layer and deepening below, in good agreement with the CARS climatology (Fig.S1a-d).

       In contrast, the ESM SST (CNRM is shown here as an example, similar results are found for IPSL and GFDL) in present conditions (2006-2015) displays a warm bias of 2-4°C typical of ESMs (Flato et al., 2013) and no clear sign of coastal
upwelling (Fig.2c). In 2091-2100, the RCM displays a coastal upwelling of  waters ~2-3°C warmer than in 2006-2015 (Fig.2d). Again the ESM SST spatial pattern in 2091-2100 (Fig.2e) resembles that of 2006-2015. Coastal upwelling is not present and a warming of ~2-3 °C is found over the main part of the domain.

**Trends of nearshore SST**

A steady warming of the surface coastal ocean is found in the three regional simulations (Fig 3a). SST increases rapidly in R-IPSL since the 2020s, reaching +4.5°C in 2100, whereas it increases since the 2030s in the other simulations, reaching +3.5°C and +2°C in R-CNRM and R-GFDL respectively. Interestingly, decadal variability can produce decades during which the SST increase is stalled (a.k.a. "warming hiatus"), *e.g.* in 2035-2045 in R-CNRM and in 2040-2060 in R-GFDL. The ESM linear trends are very similar to the RCM nearshore warming trends (Fig.3b, Table 3). Here the offset
between the three ESM SST evolutions due to the different SST bias in 2005 (between 4-6°C among the ESMs) has been corrected  in order to better compare RCM and ESM trends. As an example, the spatial structures of the R-CNRM and CNRM SST anomalies are compared (Figs.3c,d). The similarity between the two anomaly patterns is striking. Both display a maximum warming near the coasts and west of the Galapagos where upwelling occurs.

**Temporal variability of heat and momentum fluxes**

        As expected from greenhouse effect, downward longwave radiation from the ESMs increases steadily over the 21[st] century under RCP8.5 (Fig.4a). The increase is stronger in IPSL (+10%, see Table 3) and CNRM (10%) than in GFDL (7%). This induces a decrease of the surface ocean cooling associated to net longwave radiation in the RCMs (Fig.1d).  Contrasted net downward shortwave radiation trends are simulated by the ESMs (Fig.4b). Insolation decreases quasi-linearly in CNRM
(-7%) and in GFDL (-4%), however it is modulated by decadal variability in GFDL (note the slight insolation increase in 2090-2100). On the other hand, IPSL displays no trend (0%). Furthermore, alongshore wind stress, the main driver of coastal upwelling, decreases in R-CNRM (-11%) and R-IPSL (-9%) in contrast with R-GFDL (+2%) (Fig 4c). The wind stress decrease found in R-IPSL and R-CNRM is consistent with that found in CMIP3 simulations (Goubanova et al.2010; Belmadani et al. 2013).

**Coastal upwelling**

Coastal upwelling (measured as the net offshore flux, see section 2.7) decreases strongly in R-IPSL (-23%) and R-CNRM (-25%) (Fig.5a, Table 3). These downtrends are consistent with the wind stress downtrends (Fig.4c) and mainly due to the Ekman transport contribution (Fig.5c). In contrast, coastal upwelling remains stable in R-GFDL. The upwelling is modulated by decadal variability, whose amplitude can reach 5-10% of the mean value. Decadal variability may generate decades of upwelling increase (*e.g.* 2090-2100 in R-CNRM) masking the long term decrease. Upwelling decadal variability is mainly forced by variations of the onshore geostrophic transport, which on average compensates ~50% of the Ekman transport. As Ekman transport decreases over time in R-IPSL and R-CNRM, the relative contribution of the geostrophic transport increases over time. This onshore current is driven by the higher sea level in the equatorial portion of the upwelling system than in its poleward portion (Colas et al. 2008; Oerder et al., 2015). This flow is occasionally remarkably strong (*e.g.* in 2090 in R-CNRM, 2035-2040 and 2065 in R-GFDL), whereas the trends are weak.

**Subsurface temperature anomalies**

Nearshore subsurface temperature anomalies are impacted by equatorial subsurface temperature anomalies in two ways: thermocline anomalies may propagate along the equatorial and coastal wave guide (*e.g.* Echevin et al., 2011, 2014; Espinoza-Morriberón et al., 2017, 2018), and temperature anomalies may be transported eastward and poleward by the near-equatorial subsurface jets (Fig.2a; Montes et al., 2010, 2011). The latter is particularly strong during eastern Pacific El Nino events (*e.g.* Colas et al., 2008 for the 1997-1998 event). The thermal structure of the upper layer is strongly impacted by climate change in the eastern equatorial Pacific. The depth of the 20°C isotherm (hereafter D20) is used to characterize the thickness of the warm surface layer. It increases in all ESMs, at different rates (Fig.6a). The deepening is roughly linear in GFDL (+5%, Table 3) and CNRM (+26%). In contrast, it increases non-linearly in IPSL, first by ~1.5 m/decade between 2005 and 2065 , and then by ~5 m/decade between 2065 and 2100. Note that D20 is shallower in the ESMs (~30-40 m) than in observations (~52 m in WOA) and in two ocean reanalysis (~56 m in GLORYS2V1 and -58 m in SODA). A shallow thermocline is likely to be more impacted by greenhouse-induced surface warming in the model simulations than in the real ocean.

D20 coastal trends in the RCMs (Fig.6b) are roughly similar to the offshore ESM equatorial trends. The coastal deepening is moderate in R-GFDL (+12%, Table 3). In contrast, a strong linear deepening is found in R-CNRM (+101%). As in the equatorial region, the D20 deepening is non linear in R-IPSL and the thickness of the warm surface layer more than doubles (+207%). The RCM D20 values at the beginning of the century are within the range of estimated values from observations and reanalyses whereas D20 is slightly too deep in the ESMs (Fig.6c), which highlights the dynamical downscaling ability to reduce part of this systematic bias. The RCM trends are roughly in line with the ESM coastal trends. D20 deepening can be amplified (*e.g.* 207% in R-IPSL vs 126% in IPSL) or mitigated (12% in R-GFDL vs ~21% in GFDL, Fig.6c) depending on the model. Decadal variability from the equatorial region propagates to the coastal regions with little change.

We now investigate the evolution of the RCMs mixed layer. The RCM surface boundary layer thickness (hbl), determined by comparing a bulk Richardson number to a critical value (KPP parameterization; Large et al., 1994), is a good

proxy of the model mixed layer (*e.g.* Li and Fox-Kemper, 2017). The R-GFDL mixed layer in 2006-2015 is in fairly good agreement with the mixed layer depth (computed from temperature profiles) from the coarse 2°x2° gridded climatology of de Boyer Montegut et al. (2004), whereas R-IPSL and R-CNRM values are ~ 3 m shallower.

A shoaling of the mixed layer is found in all simulations (Fig.7), in line with the surface heating (Fig.4a, b) and reduced wind-driven mixing (Fig.4c). The shoaling is slightly stronger in R-IPSL and R-GFDL than in R-CNRM, possibly due to the stronger surface warming in R-IPSL (Table 3).

The near-equatorial subsurface, coastal subsurface and surface temperature linear trends of the RCMs and ESMs are compared in Figure 8. Near-equatorial subsurface trends are weakest in GFDL and strongest in IPSL, which is consistent with the stronger D20 deepening in IPSL (Fig.6a). A similar ranking from weakest (R-GFDL) to strongest warming (R-IPSL) is found for the coastal subsurface warming and coastal surface warming. The equatorial water masses are transported towards the coasts (Montes et al., 2010; Oerder et al., 2015) and the subsurface layer trends increase by  6% in R-GFDL, 23% in R-CNRM and  10% in R-IPSL with respect to the near-equatorial trends. The ESM trends are close to the RCM trends, which suggests that the nearshore subsurface warming is dominated by the eastward transport of warm near-equatorial subsurface waters both in the ESMs and RCMs. In the coastal region, the upper part of the 50-200m subsurface water volume is upwelled into the mixed layer where additional heat is deposited by the local atmospheric fluxes (Figs. 4a,b).The coastal SST trends increase with respect to the coastal subsurface anomalies (+17% in R-GFDL, +37% in R-CNRM, +44% in R-IPSL), underlining the impact of different local heat fluxes. The amplitude of the ESM SST trend is very close (<10% change) to that of the RCM for R-IPSL and R-CNRM, which is consistent with the spatial patterns of SST change shown in Figs.3c,d. Interestingly, the R-GFDL SST increase is ~20% weaker than that of GFDL.

**3.2 Biogeochemical response of the NHCS under RCP8.5 scenario**
We now investigate the impacts of regional climate change on the main biogeochemical characteristics of the NHCS, namely oxygenation, nutrients and productivity.

**OMZ trends in response to the equatorial circulation**
The suboxic ($O_2$< 5 µmol $L^{-1}$, Karstensen et al., 2008) subsurface waters found in the NHCS result from a subtle balance between the eastward and poleward transport of relatively oxygenated waters from the equatorial region into the upwelling region, the ventilation due to mesoscale circulation (Thomsen et al., 2016; Espinoza-Morriberón et al., 2019) and the local oxygen consumption due to the respiration of sinking organic matter. The eastward currents in the offshore equatorial region thus play an important role in the ventilation of the OMZ (Stramma et al., 2008; Montes et al., 2014; Cabré et al., 2015; Shigemistu et al., 2017; Espinoza-Morriberón et al., 2019). Following Cabré et al. (2015) we first evaluate the ESM eastward subsurface flow (which enters the western boundary of the RCM) at 95°W (Fig.9a). As estimates of mean velocity from ocean reanalysis range between 0.05 m $s^{-1}$ (GLORYS12V1) and 0.09 m $s^{-1}$ (SODA), the uncertainty of the eastward flow is very high. The eastward flow in R-GFDL (in 2005-2010) is ~10% weaker than in SODA. In contrast, the eastward flow is underestimated by ~50% in R-CNRM and R-IPSL with respect to SODA, probably because of a weak EUC

and/or weak SSCCs in these coarse-grid ESMs (Cabré et al., 2015). Over 2006-2100, the eastward velocity is stable (<1%, Fig.9a, Table 3) in R-CNRM and decreases weakly in R-IPSL (-9%) and in R-GFDL (-14%).

The evolution of the eastward dissolved oxygen (DO) flux at 95°W (Fig.9b) follows approximately that of the mass flux. Due to a strong increase in equatorial DO (not shown), the DO flux uptrend is strong in R-CNRM (33%, Table 4). This contrasts with the moderate decrease of the DO flux (~ -5%) in the other two simulations. Note that the DO eastward flux is ~25-30% stronger in R-IPSL than in R-CNRM at the beginning of the century. As the eastward flow in the 2°S-10°S equatorial band is stronger in R-IPSL than in R-CNRM (not shown) and the water is more oxygenated in this latitudinal band

than within 2°S-2°N (*e.g.* Figure 4 in Cabré et al., 2015), this results in a stronger DO eastward flux in R-IPSL than in R-CNRM.

We now investigate the nearshore subsurface DO concentration in a box located between 150 km and 300 km offshore, in order to take into account a sufficient number of coarse ESM grid points in the 100-200m depth range. The RCM is able to represent the cross-shore structure of the OMZ with a fair degree of realism (Figs.S1-2). The OMZ bias are weak (< 10 μmol

L$^{-1}$, Fig.S2)  below ~100m and increase near ~50-100 m, in the depth range of the oxycline/thermocline. The nearshore DO concentration in the upper part of the OMZ (between 100 and 200m, Fig.10a) in 2006-2015 is slightly higher in R-GFDL (~+20 μmol L$^{-1}$) than in the observations (~15-18 μmol L$^{-1}$) and lower in R-IPSL (~10 μmol L$^{-1}$)  and R-CNRM (~-5 μmol L$^{-1}$, see also Fig.S1).

In contrast, the ESMs strongly overestimate DO in the OMZ (Fig.10b).  The eastward flux at 95°W supplies DO to

the nearshore OMZ in greater proportions in R-GFDL than in R-IPSL and R-CNRM (Fig.9b), partly explaining the discrepancies at the beginning of the century.

The nearshore trends are very different in the three regional simulations. The DO content is virtually unchanged in R-GFDL (-3%, Table 4) and decreases slowly (-21%) in R-IPSL, whereas it increases strongly in R-CNRM (+483% ~ 30 μmol L$^{-1}$ increase). R-GFDL is also marked by a stronger multidecadal variation than the other RCMs.  The trends have the same

sign as those of the ESMs (Fig.10b), but DO changes are reduced by half in the RCMs (*e.g.* ~+60  μmol L$^{-1}$ in CNRM versus ~+30 μmol L$^{-1}$ in R-CNRM, ~-6  μmol L$^{-1}$ in IPSL  versus  ~-2.5 μmol L$^{-1}$ in R-IPSL).

The depth of the  0.5 mL L$^{-1}$ (22 μmol L$^{-1}$) DO iso-surface (hereafter named "oxycline") is often used as a proxy for the OMZ upper limit (*e.g.* Espinoza-Morriberón et al., 2019) characterizing the vertical extent of the habitat of many living species of the coastal ecosystem (Bertrand et al., 2010, 2014). As R-CNRM oxycline is quite deep (Fig.S2), we averaged its

values over a wider coastal box (0-200 km) in Figure 10c. The oxycline at the beginning of the century is well positioned in R-GFDL, and slightly shallower than the observed oxycline in R-IPSL and R-CNRM (Fig.10c). Between 2006 and 2100, the oxycline shoals slightly (less than 10 m) in R-GFDL and R-IPSL whereas it deepens of more than 100 m in R-CNRM. Similar trends are found for the "upper oxycline" defined by the 1 mL L$^{-1}$ isoline (not shown, see Table 4).

**Nitrate trends**

We now investigate the evolution of subsurface nitrate concentrations at 95°W, the western boundary of the RCM (see red line in Fig.2b). A decrease is found in all simulations. This is illustrated by the shoaling of the 21 μmol L$^{-1}$ nitrate iso-surface (Fig. 11a). The downtrends vary between strong (78% in R-CNRM) and moderate deepening (24% in R-IPSL and

26% in R-GFDL, Table 4). Nitrate depletion was also found in IPSL CMIP3 4xCO$_2$ scenario (Brochier et al., 2013). It is likely caused by a reduced nutrient delivery from the deep ocean to the upper layers of the ocean associated to enhanced thermal stratification, reduced vertical mixing and overall slowdown of the ocean circulation (*e.g.* Frölicher et al, 2010). Due to the stronger eastward flow in R-GFLD (Fig.9a), the associated nitrate eastward flux is ~50% stronger than in R-IPSL and R-CNRM (Fig. 11b). The fluxes decrease in all simulations (-27% in R-CNRM, -20% in R-IPSL, -18% in R-GFDL, Table 4, Fig.11b).

Following Espinoza-Morriberón et al. (2017), the depth of the 21 μmol L$^{-1}$ nitrate iso-surface (hereafter D21) in the coastal region is chosen as a proxy of the nearshore nitracline depth (Fig. 11c). In spite of the offshore nitracline deepening (Fig.11a) and decreasing nitrate flux (Fig.11b), the nearshore nitracline shoals in R-GFDL (-25%). In contrast, it deepens in R-IPSL (+32%) and in R-CNRM (+82%). This shows that the equatorial forcing is not always the main forcing of the evolution of the nearshore nitracline depth: whereas it seems to drive nitrate depletion in R-CNRM and R-IPSL, the maintained coastal upwelling in R-GFDL (Fig.5a) may partly compensate this effect. It is also notable that the nitracline may shoal even though coastal upwelling does not increase (*e.g.* in R-GFDL, Fig. 5a). This points to potential changes in nitrate vertical distribution, possibly due to a reduction of nitrate assimilation driven by biomass variations (see section 3.3). The ESMs and RCMs nearshore nitracline trends are consistent for CNRM and IPSL: nitracline deepens by 97% (34%) in CNRM (IPSL) and by 82% (32%) in R-CNRM (R-IPSL). In contrast, nitracline shoaling is strong in R-GFDL (-25%) and negligible in GFDL (+2%). However, note that D21 is too shallow in RCMs (~20-35 m over 2006-2015) with respect to observations (~100 m in CARS) due to an overly high nitrate concentration in subsurface layers (figure not shown). This bias was also found in previous ROMS-PISCES regional simulations of the NHCS (e.g. see also Fig.3 in Espinoza-Morriberón et al., 2017) possibly due to a lack of denitritification.

**Chlorophyll and  primary productivity annual variations**

Regional downscaling has a strong impact on the nearshore planktonic biomass. Chlorophyll is used in the following as a proxy of total phytoplankton biomass. The surface chlorophyll concentration at the beginning of the century (Fig.12a) agrees relatively well with MODIS mean chlorophyll  (~4.25 mg Chl m$^{-3}$) in R-IPSL (~4.2 mg Chl m$^{-3}$) and R-GFDL (~4.5 mg Chl m$^{-3}$) whereas it is ~30% higher in R-CNRM (~5.5 mg Chl m$^{-3}$). Note that MODIS and SeaWIFs satellite observations differ by ~1 mgChl m$^{-3}$ due to different algorithms (O'Reilly et al., 1998; Letelier and Abbott, 1996) and different time periods (cf section 2.6).  Moderate uptrends are found in R-GFDL (+12%) and R-IPSL (+17%, Table 5). The latter seems at odds with the weak nitracline deepening (<10 m between 2006 and 2100) in R-IPSL (Fig.11c). Strong multidecadal variability with almost no trend (2%) is found in R-CNRM, in spite of the marked nutricline deepening (~20 m, Fig.11c).

RCMs are able to correct the ESM inability to represent nearshore surface chlorophyll concentration (Fig.12b). Indeed, ESM surface chlorophyll range between ~0.6-0.7 mgChl m$^{-3}$ (GFDL)  and ~0.01-0.1 mgChl m$^{-3}$ (CNRM), almost an order of magnitude smaller than observed values. The ESM trends display very contrasted patterns (Fig.12b). Surface chlorophyll concentration  decreases in all cases, with negative trends between -11% and -104%,  a behavior not simulated in the RCMs.

The total chlorophyll content, depth-integrated over 0-500m (which includes the euphotic layer) (Fig.12c), displays

weak uptrends in R-IPSL (+2%, Table 5) and R-GFDL (+3%) and a moderate decrease in R-CNRM (-5%). Note also the very marked multidecadal variability in R-CNRM. In contrast, weak downtrends (-3%) are found in two of the ESMs (IPSL and GFDL, Fig.12d). Note that the R-CNRM downtrend (-5%) is weaker (-8%) with respect to CNRM (-32%).

The different evolution of the RCM surface and total chlorophyll content implies that the vertical distribution of phytoplankton biomass is modified in the long term. The vertical and cross-shore structure of seasonal chlorophyll trends

indicates that both R-GFDL and R-IPSL simulate a chlorophyll increase in the mixed layer near the coast, and a decrease below (Figs.13a-c). Interestingly, this suggest that total biomass changes cannot be monitored using satellite measurements, as the subsurface plankton depletion cannot be observed. The seasonal trends in R-GFDL and R-IPSL are consistent with a shoaling of the mixed layer (Fig.7), which reduces light limitation of phytoplankton growth (*e.g.* Echevin et al., 2008; Espinoza-Morriberón et al., 2017) and increases surface primary productivity in summer and winter. In contrast, the R-

CNRM trend in the mixed layer is negative in summer. This is likely caused by the strong deepening of the nitracline in R-CNRM (Fig.11c) and the seasonality of the wind-driven upwelling. As the upward flow is weaker in summer, the upwelling of less rich waters into the mixed layer may trigger a nutrient limitation of phytoplankton growth. On the other hand, as the upward flow remains strong during winter, nutrient limitation does not occur. Light limitation of phytoplankton growth reduces because of the shoaling of the mixed layer, enhancing phytoplankton growth (as in the two other RCMs). Moreover,

visual correlation between decadal variability of the chlorophyll content and nitracline depth in R-CNRM (*e.g.* the oscillations in 2070-2100 in Fig.11c and Fig.12c) also suggests that nitrate limitation of phytoplankton growth may play a role.

To further investigate the drivers of the surface chlorophyll trends, RCM and ESM primary productivity (PP) trends are shown in Fig.14. RCM PP surface trends are weak (between -2% and +7%). In particular, the weak trend in R-IPSL (-2%) is

at odds with the surface chlorophyll increase (+17%, Fig.12a). In all RCMs, PP is strongly impacted by decadal variability as a consequence of upwelling (Fig.5a) and nitracline depth variability (Fig.11c). These surface trends contrast with the more pronounced ESM PP trends, in particular for IPSL (-25%) and CNRM (-113%). However, one may question the meaning of the ESM PP trends associated with very weak (an irrealistic) ESM chlorophyll concentrations (Figs.12b,d). The RCM depth-integrated PP trends are consistent with to those of surface PP but differ from the ESMs, especially for R-CNRM (-7%) and

CNRM (-66%).

Overall, the contrasted trends found in the RCMs and ESMs, even when a similar biogeochemical model is used (*e.g.* PISCES in IPSL and CNRM), illustrate the necessity to regionally downscale ESM variability to reduce systematic bias and better represent local processes impacting on productivity.

**Zooplankton biomass variations**

The two zooplankton groups represented by RCMs are aggregated in a single group to allow a comparison with the ESMs. In contrast with surface phytoplankton, the order of magnitude of surface zooplankton biomass is comparable in EMSs and RCMs, with the exception of CNRM in which zooplankton concentrations are very weak. Besides, RCM surface zooplankton also displays a different evolution than RCM phytoplankton. First, multidecadal variability is quite strong and

trends are weak. Zooplankton slightly accumulates in R-GFDL (+4%, Fig.15a, Table 5), in line with phytoplankton (+12%,

Fig.12a), suggesting the possibility of a grazing increase. In contrast, surface zooplankton displays no trend in R-IPSL in spite of a marked surface phytoplankton increase (+17%). These weak surface zooplankton trends contrast with the stronger ESM downtrends (from -15% (GFDL) to -98% (CNRM), Fig.15b).

Depth-integrated zooplankton biomass decreases moderately in all RCMs, from -5% (R-GFDL) to -15% (R-IPSL) (Fig.15c). The GFDL and IPSL depth-integrated zooplankton downtrends are relatively close to the RCM downtrends. CNRM stands out as atypical with a decrease of half of its zooplankton biomass, while the decrease in R-CNRM is moderate (-11%). The spatial structure of the trends varies significantly over the vertical and in the cross-shore direction (Figs.16a-c). The accumulation of zooplankton in R-IPSL and R-CNRM near the coast is consistent with a reduction of the offshore advection due to Ekman transport (Fig.5c). As for chlorophyll (Fig.13), the zooplankton decrease below 10 m depth suggests that monitoring of zooplankton must be carried out in the surface layer and below to measure long-term trends.

## 4 Summary and discussion

### 4.1 Summary of the main results

The dynamical downscaling of the ocean circulation and ecosystem functioning for three ESMs is performed in the NHCS for the strongly warming, so-called "worst-case" RCP8.5 climate scenario. The RCM simulations all show an intense warming of the surface layer within 100 km from the Peruvian coasts, reaching between +2°C and +4.5°C in 2100. We can speculate that the nearshore surface warming is closely associated with a subsurface warming in the near-equatorial region (95°W, 2°N-10°S) which propagates into the NHCS. The coastal warming is weakest when the wind-driven upwelling is maintained (*e.g.* in R-GFDL), and strongest when it is reduced (*e.g.* in R-IPSL and R-CNRM, see also Echevin et al., 2012; Oerder et al., 2015). The coastal warming found in the RCMs is close to that found in the ESMs, but surface and subsurface temperature mean biases (for the period 2006-2015) are greatly reduced in the RCMs.

Biogeochemical trends from the RCMs and ESMs are compared. Two of the three RCMs display a weak decrease of the near-equatorial (95°W, 2°N-10°S) eastward oxygen flux into the NHCS, associated with a moderate slowdown of the eastward equatorial circulation and weak changes in oxygen concentrations in the equatorial region. Consequently, a relatively weak deoxygenation occurs in the nearshore region. This contrasts with the third RCM, in which the near-equatorial region becomes very oxygenated, which triggers a strong oxygenation of the OMZ.

Nutrient supply from the near-equatorial region to the NHCS decreases in all RCMs due to progressive nitrate depletion of equatorial waters and to decreasing eastward flux. This drives a deepening of the nearshore nitracline in two of the RCMs, and a shoaling in the third RCM in which wind-driven coastal upwelling is maintained.

Chlorophyll concentration displays contrasted coastal trends. First, in all RCMs, surface chlorophyll does not decrease, in contrast with ESM downtrends (from -11% to -104%). Surface chlorophyll increases (>10%) in two RCMs, while the total chlorophyll biomass remains stable, indicating an enhanced vertical stratification of phytoplankton in the surface layer in 2100. Total phytoplanktonic biomass (*i.e* integrated over the water column) in the coastal zone remains relatively stable in spite of a slightly decreasing primary productivity driven by a weakening upwelling (in two RCMs) and a deepening nutricline (in two RCMs). This counterintuitive evolution of surface phytoplankton could be partly driven by the reduced offshore transport (related to coastal upwelling) which allows floating organisms to accumulate in the coastal band.

Reduced offshore transport may also induce a greater residence time of phytoplankton in the coastal area hence a stronger prey availability favoring grazing and a larger zooplankton biomass. However, the total zooplankton biomass tends to decrease in all RCMs, which shows that complex nonlinear effects (*e.g.* temperature and predator-prey relations) drive plankton trends. Note that RCM zooplankton downtrends can be weaker than the ESM downtrends used to drive fish global models (*e.g.* Tittensor et al. , 2018). In the following subsections we discuss in more details the surface temperature trends, the near-equatorial conditions impacting the NHCS and the impact of the downscaling on the plankton trends.

**4.1 Selection of the ESMs**

The choice of which ESMs to downscale has been justified on the basis of the comparison of the ESMs historical simulations to climatological observations. We are aware that these evaluations do not necessarily correspond to how well a model may capture the response to future climate forcing. The "Emergent Constraints" approach has been offered as a relevant method for evaluating climate models (*e.g.* Hall et al. 2019). In this approach, a statistical relation (F) between a present state variable (X) and a future state variable (Y) is derived (Y=F(X)) using an ESM ensemble, regardless of ESM bias. The relation is then used to derive a future response using the best knowledge of the present state (X_obs) using Y=F(X_obs). Following such an approach would have been useful to select the ESM models that fit best with the relation F. However, as we are interested in several variables (thermal stratification, upwelling, productivity, OMZ), this would necessitate finding distinct "emergent constraints" for these variables, and thus possibly selecting different ESMs for each constraint, which may be intricate. Such an approach is however promising and should be envisaged in future work.

**4.2 SST warming**

Enhanced surface heat fluxes and coastal upwelling of offshore-warmed source waters appear to be the main drivers of the nearshore SST evolution. The strongest nearshore warming (+4.5°C in 2100) found in R-IPSL likely results from the superposition of four effects: (i) a stronger warming of subsurface waters in the near-equatorial region subsequently transported towards the coastal region, (ii) a reduced cooling due to a decreasing coastal upwelling driven by the wind relaxation, (iii) a stable shortwave flux and (iv) an increasing downward longwave flux due to the greenhouse effect. Moreover, IPSL-CM5 ranks among the high-sensitivity climate models of CMIP5 due to a large positive low-level clouds feedback (Brient and Bony, 2013). The weaker surface warming in R-CNRM (+3.5°C in 2100) may be mitigated by the weaker insolation. Last, the weakest warming in R-GFDL (+2°C in 2100) can be explained by (i) the weakest offshore subsurface temperature anomalies, (ii) the strongest wind-driven coastal upwelling (which brings deeper colder waters to the surface layer) and (iii) the weakest greenhouse forcing. As upwelling-favorable winds are more likely to decrease than to increase in low-latitude EBUS such as the Peruvian system (Goubanova et al., 2011; Belmadani et al., 2014; Rykacsewski et al., 2015), an upwelling reduction and strong SST warming appears to be the most robust projection. However, a rigorous estimate of the forcing terms in the nearshore heat budget would necessitate the online computation of each term (*e.g.* Echevin et al., 2018).

Warmer surface waters may have severe consequences on the functioning of the Humboldt current ecosystem as a whole (Doney, 2006; Doney et al., 2012). For instance, in spite of the broad temperature range of small pelagic fish species

(*e.g.* anchovy, sardine or jack mackerel) habitat (*e.g.* Gutierrez et al., 2008), the temperature anomalies associated with El Niño events may drive the NHCS into conditions detrimental for pelagic recruitment. Moreover, previous modelling studies based on the RCP8.5 scenario suggest that Peruvian fisheries will be impacted by the poleward migration of exploited species to encounter cooler waters (*e.g.* Cheung et al., 2018).

## 4.3 Near-equatorial eastward flow and OMZ variability

Eastward EUC and SSCCs are supposed to be strong drivers of OMZ variability as they transport relatively oxygenated equatorial waters into the OMZ (Cabré et al., 2015; Shigemitsu et al., 2017; Montes et al., 2014; Espinoza-Morriberón, 2019; Busecke et al., 2019). This is in line with our results: in all  RCMs, the DO trend in the OMZ is consistent with the trend of the offshore DO eastward flux. The EUC is supposed to be mainly forced by the zonal pressure gradient across the equatorial Pacific, associated to the trade winds and the Walker circulation (hereafter WC; Stommel, 1960). However, most of the CMIP5 climate models fail to reproduce the WC intensification observed in the recent period (1980-2010) (*e.g.* Kociuba and Power, 2015). Furthermore, the EUC decrease in the eastern equatorial Pacific in GFDL and in IPSL (respectively -26% and -22% decrease between 2005 and 2100 for the mean velocity between 2°N and 2°S, 95°W, 50-200 m depth, Figure not shown) is not consistent with the WC trends reported in Kociuba and Power (2015). Note also that EUC trends vary significantly across the equatorial Pacific (Drenkard and Karnauskas, 2014). EUC dynamics are also likely sensitive to stratification changes in the equatorial thermocline (McCreary, 1981). In brief, to our knowledge, the mechanisms driving EUC long-term variability in the eastern equatorial Pacific remain to be investigated.

SSCCs long-term variability, which contributes to the NHCS trends (*e.g.* Montes et al., 2014), is also unknown. At basin scale, the primary SSCC (near 4°-6°S at 90°W) is supposed to be forced partly by trade winds and alongshore winds in the NHCS, by mass exchange between the Pacific basin and the Indian ocean, and by surface heating in the tropics (McCreary et al., 2002; Furue et al., 2007). The problem is that SSCCs are not resolved in CMIP5 models due to coarse resolution (*e.g.* see Fig.4 in Cabré et al., 2015). Last, the observed deoxygenation of water masses in equatorial regions (Stramma et al;, 2008) is underestimated in global models (Oschlies et al., 2018). These uncertainties imply that the ventilation of the NHCS OMZ by the eastward jets may be difficult to project using CMIP5 ESMs.

In order to investigate further the impact of the ESM oxygen conditions on the RCM results, we conducted a series of sensitivity simulations (called R-GCM') using climatological seasonally-varying WOA DO concentrations at the regional model open boundaries. Boundary conditions for all the other biogeochemical variables are unchanged with respect to the reference simulations (RCM) (note that we are aware that this simplification introduces inconsistencies in the biogeochemical properties of the water masses. However the results are worth reporting). As expected, the DO eastward flux (Fig.17a) now follows roughly the mass flux evolution (Fig.9a) and decreases weakly in each simulation. The huge nearshore DO trend previously found in R-CNRM (+483%, Fig.10b) is now much weaker in R-CNRM' (+36%) and of a comparable order of magnitude as the other RCM's (Fig.17b). Furthermore, the marked decrease of the eastward DO flux in R-GFDL' appears to drive a strong nearshore DO decrease. This confirms that strong changes in the near-equatorial eastward ventilation flux impact the OMZ, in line with previous studies (*e.g.* Shigemitsu et al., 2017). However, ventilation of the OMZ by this mechanism is not the only driver of oxygen variability. Indeed, nearshore deoxygenation can vary (it is slightly more intense

in R-IPSL than in R-GFDL, Fig.10a) in spite of rather similar decrease of the near-equatorial DO eastward fluxes, possibly
owing to different local physical and biogeochemical processes (and thresholds). Computing a rigorous DO budget in the
coastal region is needed to investigate in more details the local processes at stake.

### 4.4 Plankton trends

A stable and, in one case, increasing concentration of chlorophyll are found in the surface layer (0-5m), in spite of
primary production decrease (*e.g.* in R-CNRM and R-IPSL, Fig.14). Several mechanisms could contribute to partly
compensate the PP decrease.

The shoaling of the mixed layer may constrain phytoplankton vertically and increase surface concentration. The
increased temperature in the near-surface layer (0-50 m depth) induces a faster growth rate of phytoplankton cells (Eppley,
1972). Furthermore, the decrease of upwelling and offshore export (Fig.5) may concentrate more biomass in the coastal
region and contribute to the phytoplankton persistence in R-IPSL and R-CNRM. However, performing a budget of
phytoplankton in the model would be needed to estimate precisely the relative contribution of each process , but this is
beyond the scope of the present study.

Examination of RCM zooplankton biomass shows weak trends (0-4%) in the surface layer and weak downtrends
(between -5% and -15%) for total biomass (Fig.15). R-IPSL and R-GFDL zooplankton biomass decrease faster than
phytoplankton, which corresponds to a trophic attenuation of the transfer of biomass to upper levels. A similar attenuation has
been found in regional simulations of the Benguela upwelling system under the IPCC-AR4 A1B scenario (corresponding to
the more moderate RCP6.0 scenario; Chust et al., 2014). The RCM zooplankton trends also contrast with the ESM
downtrends. These discrepancies can be attributed to local physical processes (transport and mixing associated to the
mesoscale) not represented in the ESMs, but also partly to the use of an earlier version of the ecosystem model (PISCES) run
with a set of biogeochemical parameters adapted for the NHCS (see Table 1 in Echevin et al., 2014). The stronger total
zooplankton biomass downtrends in R-CNRM and R-IPSL suggest a strong impact of the temperature increase, possibly due
to the higher zooplankton mortality in a warmer environment. However, the model's microzooplankton and mesozooplankton
result from a nonlinear interplay of temperature and predation/mortality effects. Further interpretation of these trends would
require dedicated sensitivity experiments and performing a zooplankton budget. This is beyond the scope of the present study
which aims to present an overview of the main low trophic level trends.

### *5* Conclusions and perspectives

Regional downscaling of three coarse-grid ESMs is performed in the NHCS over the 21[st] century so-called "worst-
case" RCP8.5 climate scenario using a high-resolution regional coupled biodynamical model. The downscaling procedure
allows to correct ESM bias. All regional simulations reproduce an intense warming (2-4.5°C) of the surface layer within 100
km from the Peru coasts. The surface warming is strongest when the subsurface equatorial warming is strong and the wind-
driven coastal upwelling weakens in the future.  Downscaled trends are consistent with those obtained from the  ESMs.

The biogeochemical impacts of climate change are more contrasted among RCMs and ESMs. A slowdown of the
eastward near-equatorial circulation may reduce the ventilation of the NHCS and induce a nearshore deoxygenation trend.

However the long-term variability of oxygen content of equatorial water masses also impacts the nearshore oxygen trends. As observed deoxygenation trends in the eastern equatorial Pacific are not well reproduced by ESMs (Stramma et al., 2008, 2012) and CMIP5 ESM systematic biases are strong in this region (Cabré et al., 2015; Oschlies et al., 2018), these shortcomings limit the predictability of downscaled oxygen trends in the NHCS. One important conclusion of our study is that reducing the biases in oxygen concentration and zonal circulation trends in the eastern Equatorial Pacific ocean is crucial

to project the future evolution of the NHCS oxygen minimum zone.

Downscaled surface chlorophyll in the coastal region does not decrease, in contrast with the signal projected by the ESMs. In two RCMs, the surface chlorophyll remains high in the coastal region. We can speculate that this happens for two reasons: the enhanced thermal stratification due to the warming may alleviate light limitation and vertical dilution, and the reduction of wind-driven offshore transport may allow plankton to accumulate near the coast. These processes could partly

compensate the reduction of primary productivity due to a deeper nitracline and reduced wind-driven coastal upwelling. Downscaled zooplankton downtrends are also relatively weak (between -5% and -15%) but appear to strengthen when the warming is stronger. In all RCMs, downscaled plankton trends differ markedly from those simulated by ESMs, in particular in the surface layer  (0-5m), which illustrates the strong impact of the regional dynamical downscaling. This also underlines the necessity to interpret ESM biomass-based regional projections of fisheries (*e.g.* FISHMIP, Tittensor et al., 2018) with

great caution.

As previous works point to a relaxation of upwelling-favorable wind conditions in the NHCS (*e.g.* Belmadani et al., 2014), dynamically downscaled wind projections as well as more realistic large scale dynamical and biogeochemical conditions in the near-equatorial regions are needed to improve the robustness of our results in future studies. Furthermore, many aspects of the regional impact of climate change have not been explored, such as for example interannual variability

associated with ENSO in a warmer NHCS or the acidification of coastal waters. These impacts will be addressed in future studies.

**Acknowledgements**:

Earth System Model output were downloaded from the ESGF website (https://esgf-data.dkrz.de/search/cmip5-dkrz/) and

635 from the ciclad server at the Institut Pierre-Simon Laplace (IPSL). Regional simulations were performed on the ADA and jean-Zay supercomputers at the Institut du Développement et des Ressources en Informatique (DARI projects n° A0050101140 and A0060101140) and on the IMARPE cluster. V. Echevin , F. Colas and M. Gévaudan were funded by IRD (Institut de Recherche pour le Développement), in particular by the LMI DISCOH program. The Inter-American Development Bank (IDB) is acknowledged for funding the IMARPE cluster through project "Adaptación al Cambio

Climático del Sector Pesquero y del Ecosistema Marino Costero del Perú". J. Ramos is acknowledged for for administrating the IMARPE cluster. R. Soto is acknowledged for downloading the Earth System Model output. We thank Dr. M.J. Jacox and an anonymous reviewer for their constructive comments.

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

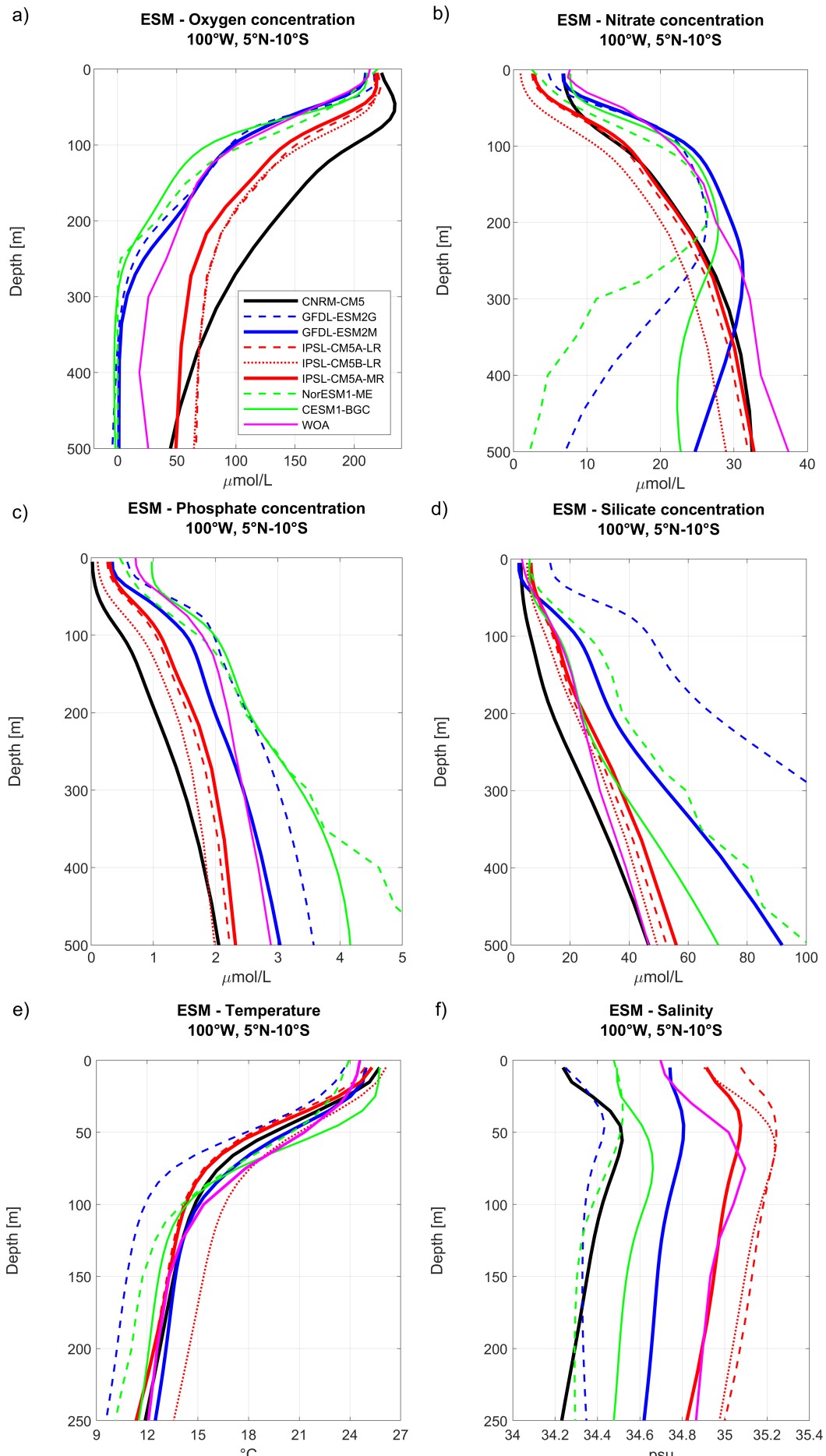

**Figure 1:** Vertical profiles of (a) oxygen, (b) nitrate, (c) phosphate, (d) silicate concentrations, (e) temperature and (f) salinity in the Eastern Equatorial Pacific Ocean for 8 Earth System Models (ESMs: CNRM-CM5 (black line), GFDL-ESM2M (blue line) ,GFDL-ESM2G (blue dashed line), IPSL-CM5A-MR (red line), IPSL-CM5A-LR (red dashed line), IPSL-CM5B-LR (red dotted line), CESM1-BGC (green line), Nor-ESM1-ME (green dashed line). All values are averaged along 100°W between 5°N and 10°S. The three selected models are shown in thick colored lines. WOA observations are marked by magenta lines. ESMs biogeochemical variables are averaged on the 1981-2005 period, while ESMs temperature and salinity are averaged on the 1950-2005 period.

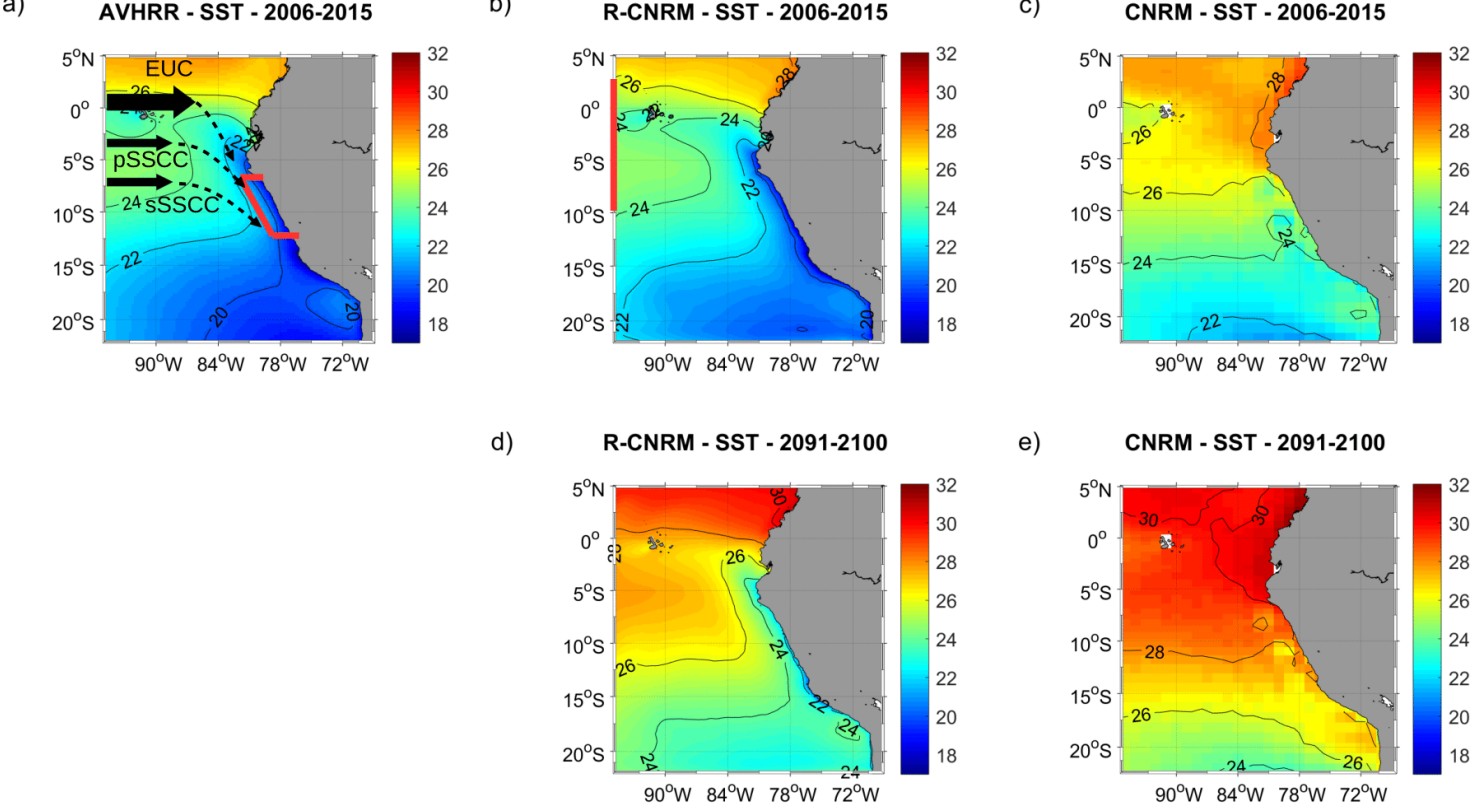

**Figure 2:** Annual mean SST (in °C) for (a) AVHRR surface observations (2006-2015), (b) R-CNRM (Control period, 2006-2015), (c) CNRM (2006-2015), (d) R-CNRM (RCP8.5, 2091-2100), (e) CNRM (RCP8.5, 2091-2100). The red box in (a) marks the coastal box in which surface and subsurface variables are averaged (see methodology section 2.8), and the red line in (b) marks the 95°W offshore section. Subsurface eastward equatorial currents (Equatorial Under Current (EUC) primary and secondary subsurface counter currents (pSSCC and sSSCC)) are sketched in (a).

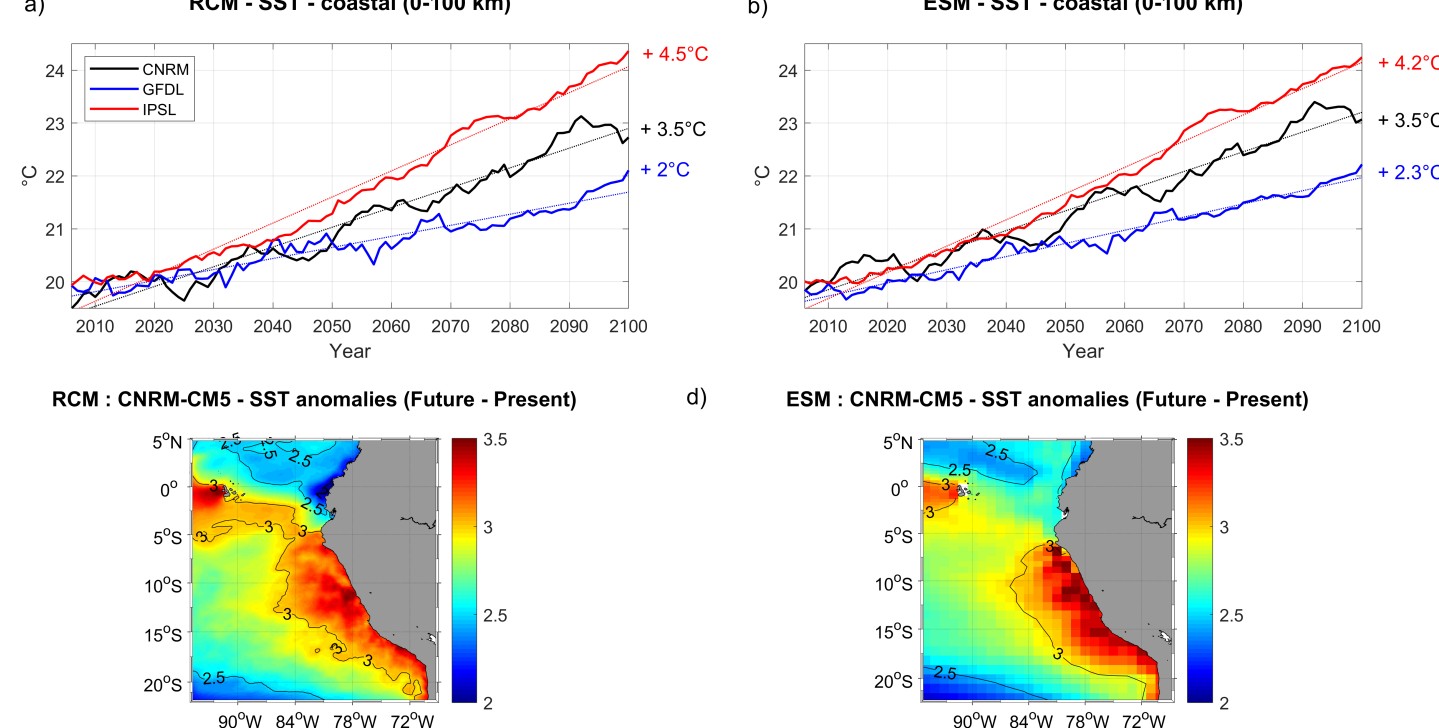

**Figure 3:** Coastal SST (in °C) in (a) the RCMs and (b) the ESMs (CNRM in black, GFDL in blue, IPSL in red). All fields are averaged in a coastal box (see Figure 2a) and annual mean time series are filtered using a 10-year moving average. SST biases are removed from ESM, to compare with RCM (respectively 5°C, 6°C and 4.5°C for CNRM, GFDL and IPSL). Dotted lines indicate linear trends and percentage values indicate the change between 2006 and 2100 with respect to present conditions using the linear trend values (i.e. 100.(X (t=2100)-X(t=2006))/X(t=2006) where X(t) is the linear trend). (c) R-CNRM SST anomaly (2091-2100 average minus 2006-2015) and (d) CNRM SST anomaly (2091-2100 average minus 2006-2015).

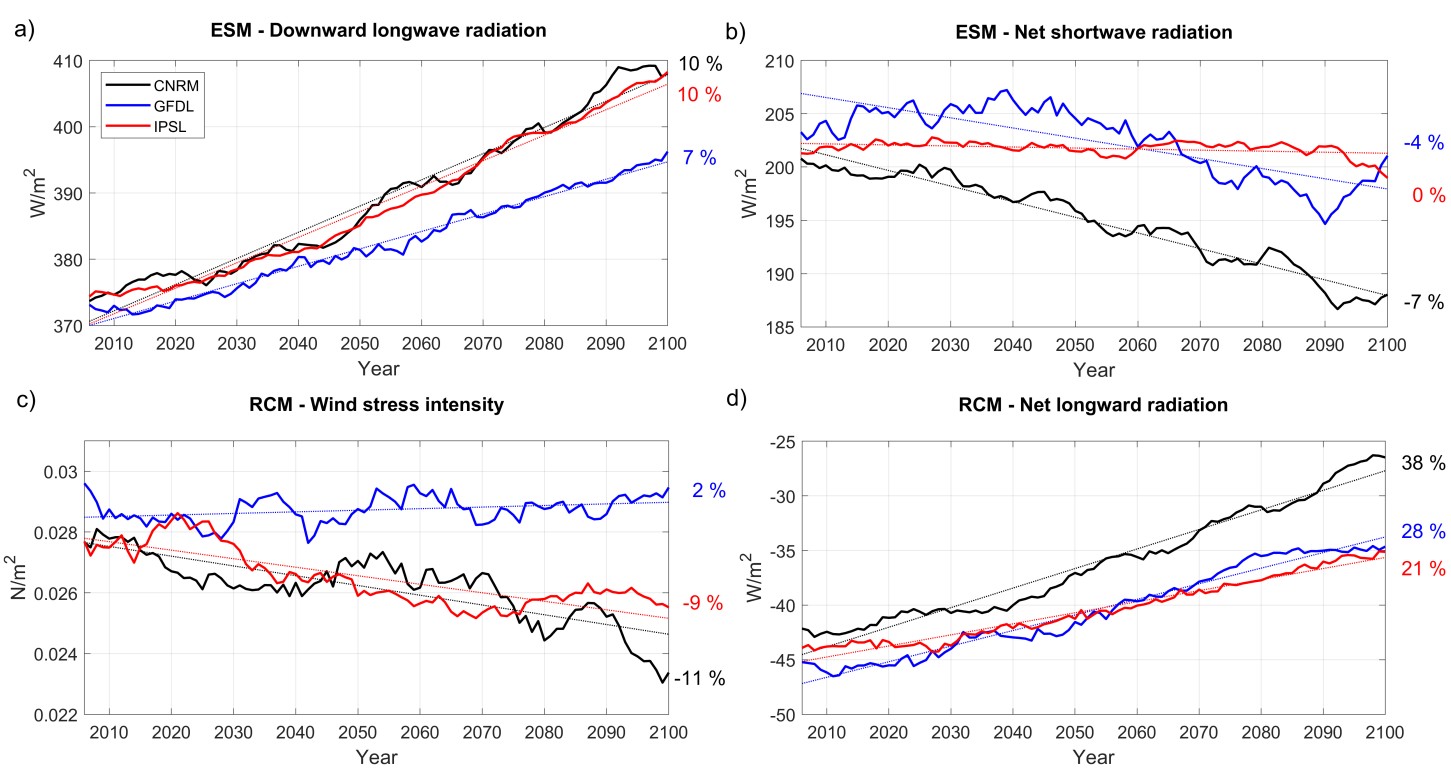

**Figure 4:** (a) ESM Downward longwave radiation (in W.m$^{-2}$, positive downward), (b) ESM net shortwave radiation (in W.m$^{-2}$, positive downward), (c) RCM wind stress intensity (in N.m$^{-2}$) and (d) RCM net longwave radiation (in W.m$^{-2}$, positive downward), for the three RCMs (CNRM in black, GFDL in blue, IPSL in red). All fields are averaged in a coastal box (see Figure 2a) and annual mean time series are filtered using a 10-year moving average.

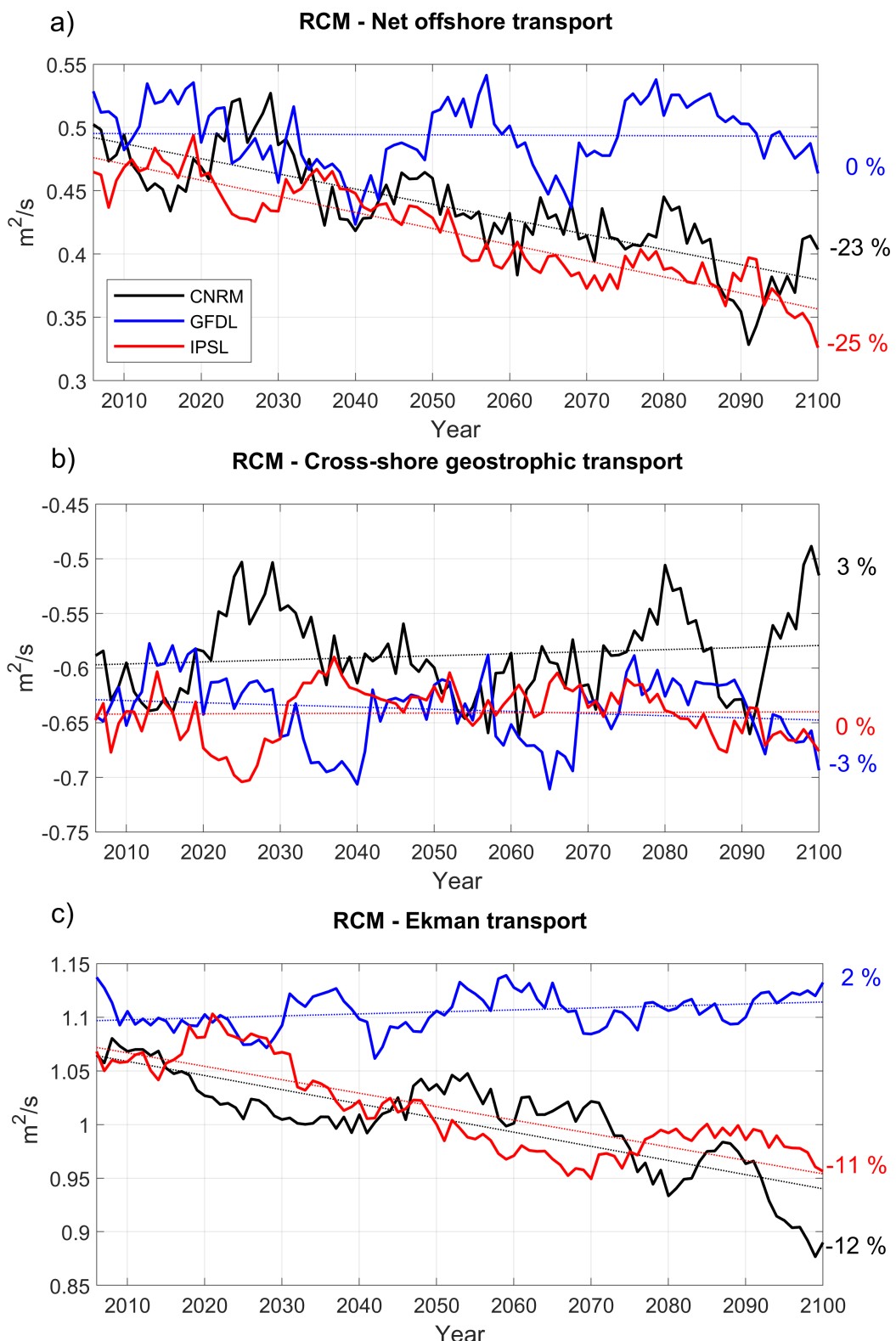

**Figure 5:** (a) Net offshore transport (in $m^2.s^{-1}$, positive offshoreward), vertically averaged in the Ekman layer, (b) cross-shore geostrophic transport compensating the wind-driven upwelling (in $m^2.s^{-1}$) and (c) Ekman transport (in $m^{-2}.s^{-1}$). All fields are averaged in a coastal box (see Figure 2a) for the three RCMs (CNRM in black, GFDL in blue, IPSL in red). Annual mean time series are filtered using a 10-year moving average.

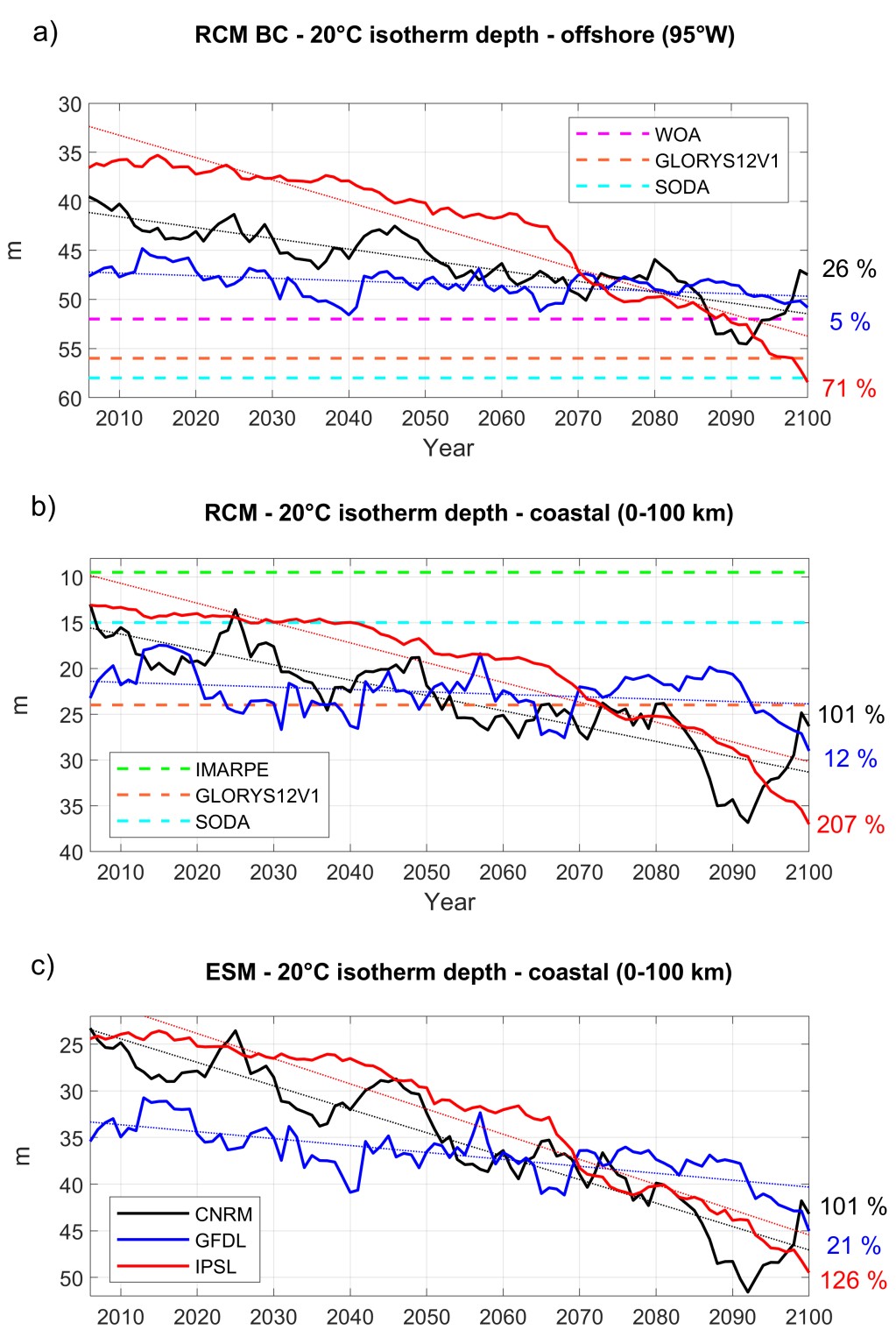

**Figure 6:** Depth of 20°C isotherm (D20, in meters) (a) at 95°W, averaged between 2°N-10°S (see red vertical line in Fig.2b), (b) in the coastal box for the three RCMs and (c) for the 3 ESMs (CNRM in black, GFDL in blue, IPSL in red). The timeseries are filtered using a 10-year moving average. Climatological D20 from WOA (dashed magenta line) and two reanalyses (dashed orange line for GLORYS12V1 (1993-2017) , dashed cyan line for SODA (1992-2000)) are also shown in (a). D20 from IMARPE climatology is marked by a dashed green line in (b). Annual mean time series are filtered using a 10-year moving average.

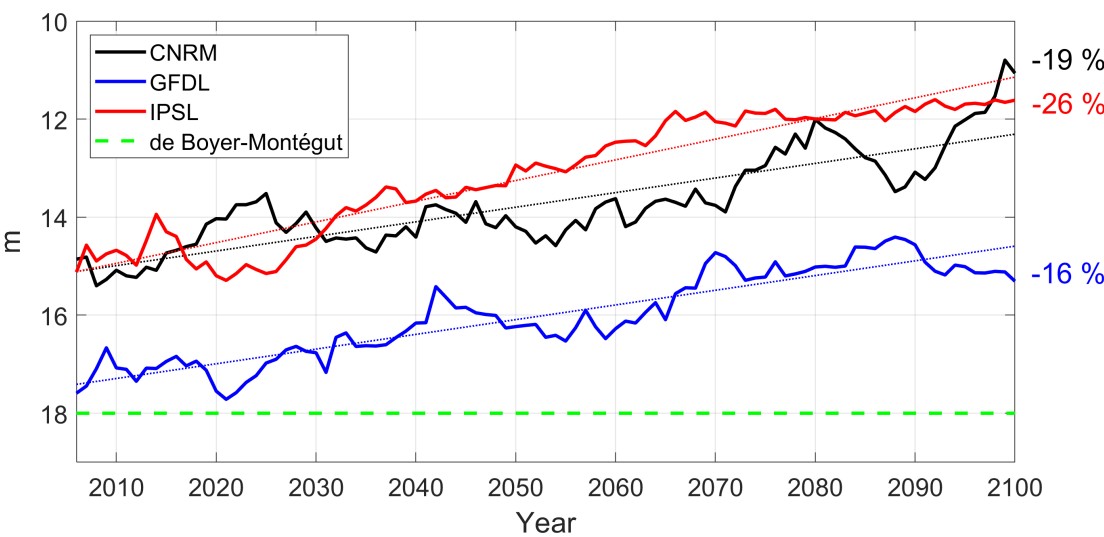

**Figure 7:** RCM mixed layer depth (in meters) in the RCMs (CNRM in black, GFDL in blue, IPSL in red). Annual mean time series are filtered using a 10-year moving average. The climatological value derived from the De Boyer Montégut et al. (2004) climatology is marked by a dashed green line.

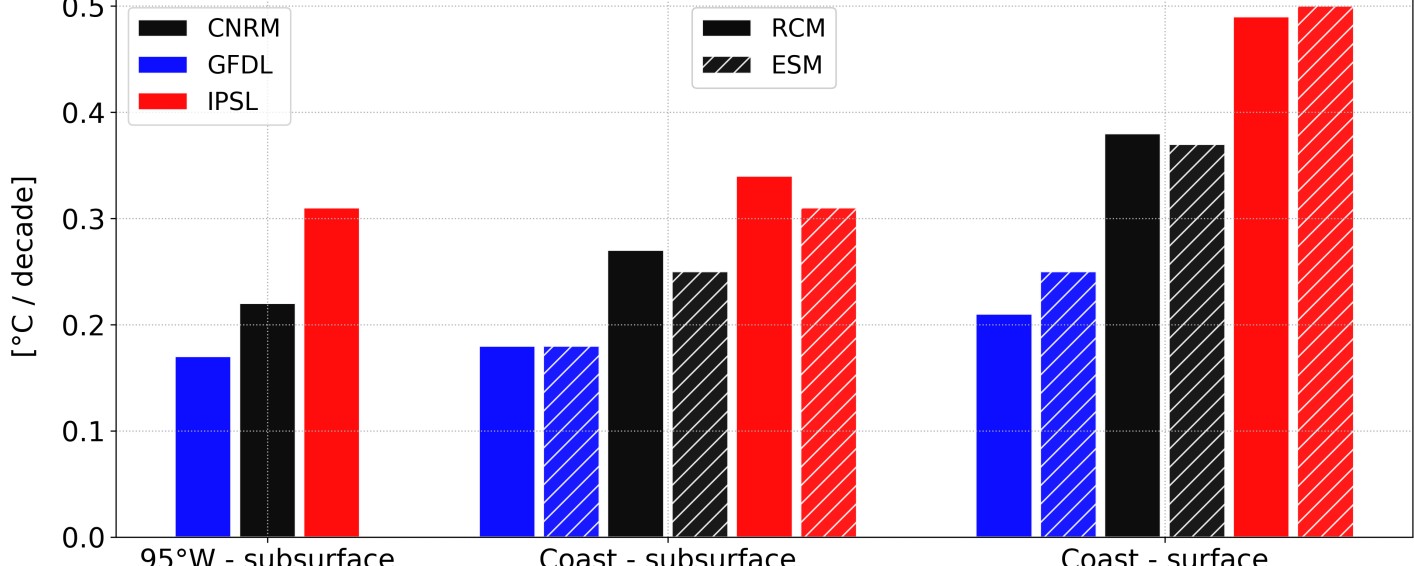

**Figure 8:** Depth-averaged RCM and ESM temperature linear trends between 2006 and 2100 (in °C/decade) in the equatorial region (95°W, 2°N-10°S, 50-200 m, left), in the coastal region (center) and in surface layer (0-5 m, right). CNRM, GFDL and IPSL trends are shown in black, blue and red, respectively, and ESM trends are shown in hatched.

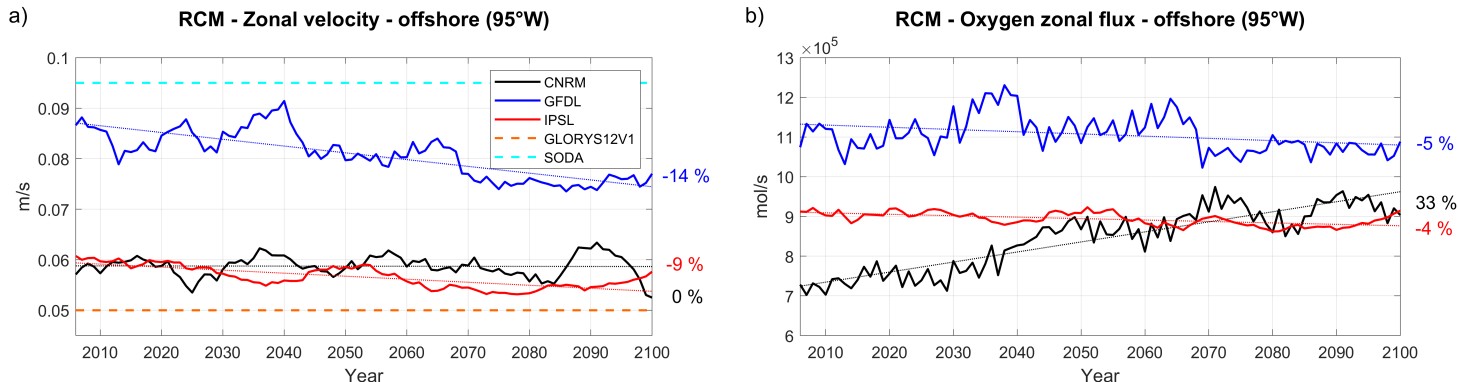

**Figure 9:** (a) zonal velocity (in m.s$^{-1}$) and (b) oxygen zonal flux (in mol.s$^{-1}$) in the eastern equatorial Pacific at 95°W, averaged between 2°N-10°S and 50m-200m, for the three RCMs (CNRM in black, GFDL in blue, IPSL in red). The timeseries are filtered using a 10-year moving average. Mean values from GLORYS12V1 and SODA reanalyses are marked in (a) by a dashed orange line and a dashed cyan line respectively.

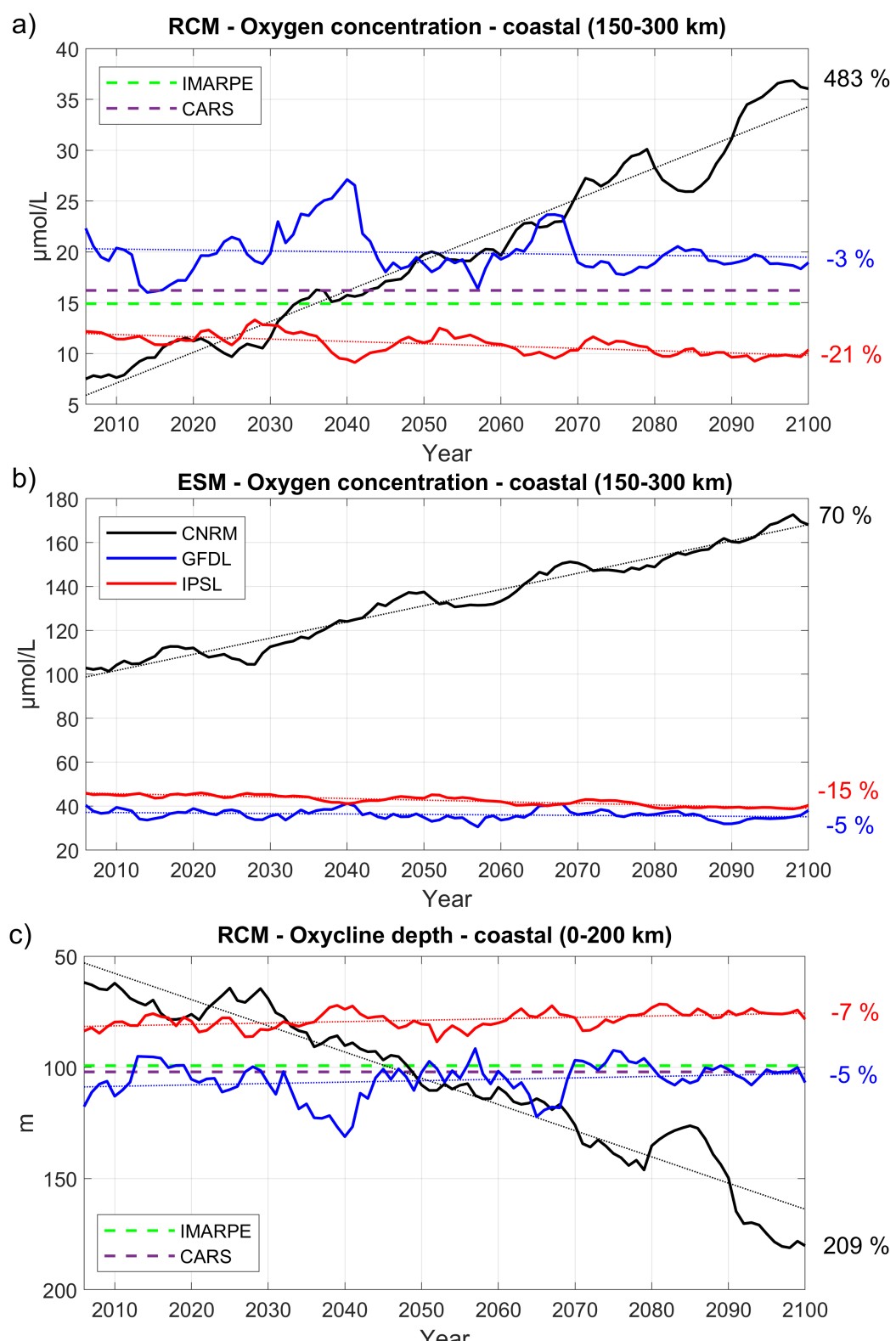

**Figure 10:** Oxygen concentration (in $\mu$mol.L$^{-1}$) averaged between 100 and 200 meters depth in coastal box located between 150 and 300 km from coast for (a) RCM and (b) ESM, (c) depth of the oxycline (0.5 mL.L$^{-1}$ $\sim$ 22 $\mu$mol.L$^{-1}$) isosurface averaged in 200 km-wide coastal box for the three RCMs (CNRM in black, GFDL in blue, IPSL in red). The timeseries are filtered using a 10-year moving average. IMARPE (dashed green line) and CARS (dashed purple line) climatological values are also shown.

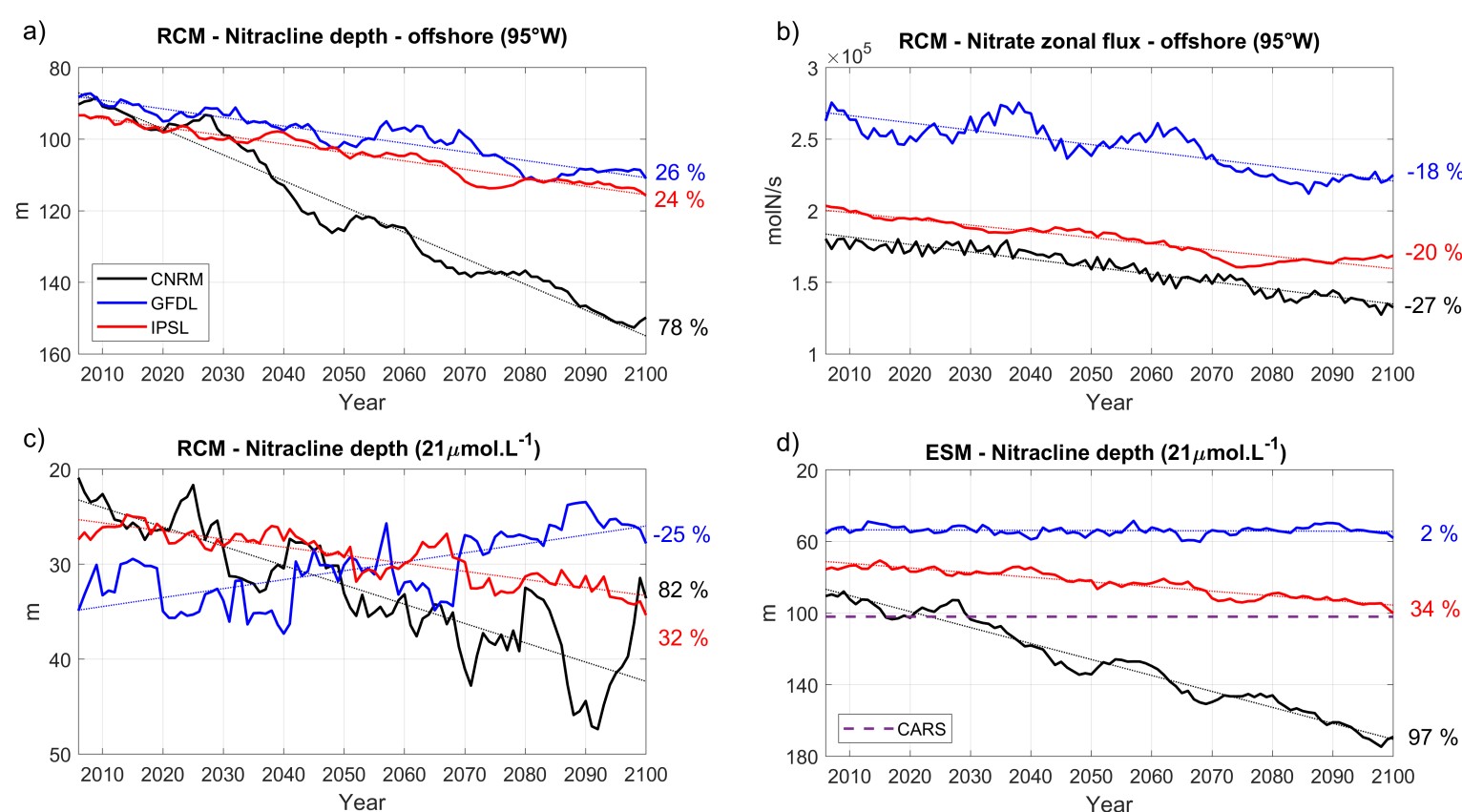

**Figure 11:** (a) Nitracline depth (i.e. depth of the nitrate 21 $\mu$mol.L$^{-1}$ isosurface) at 95°W (averaged between 2°N-10°S), (b) nitrate eastward flux (in mol.s$^{-1}$) at 95°W (averaged between 2°N-10°S, 50-200 m depth) (c) nitracline depth (i.e. depth of the nitrate 21 $\mu$mol.L$^{-1}$ isosurface) averaged in 100km-wide coastal box, for the three RCMs (CNRM in black, GFDL in blue, IPSL in red) and (d) for the ESMs. The timeseries are filtered using a 10-year moving average. CARS (dashed purple line) climatological values is shown in (d).

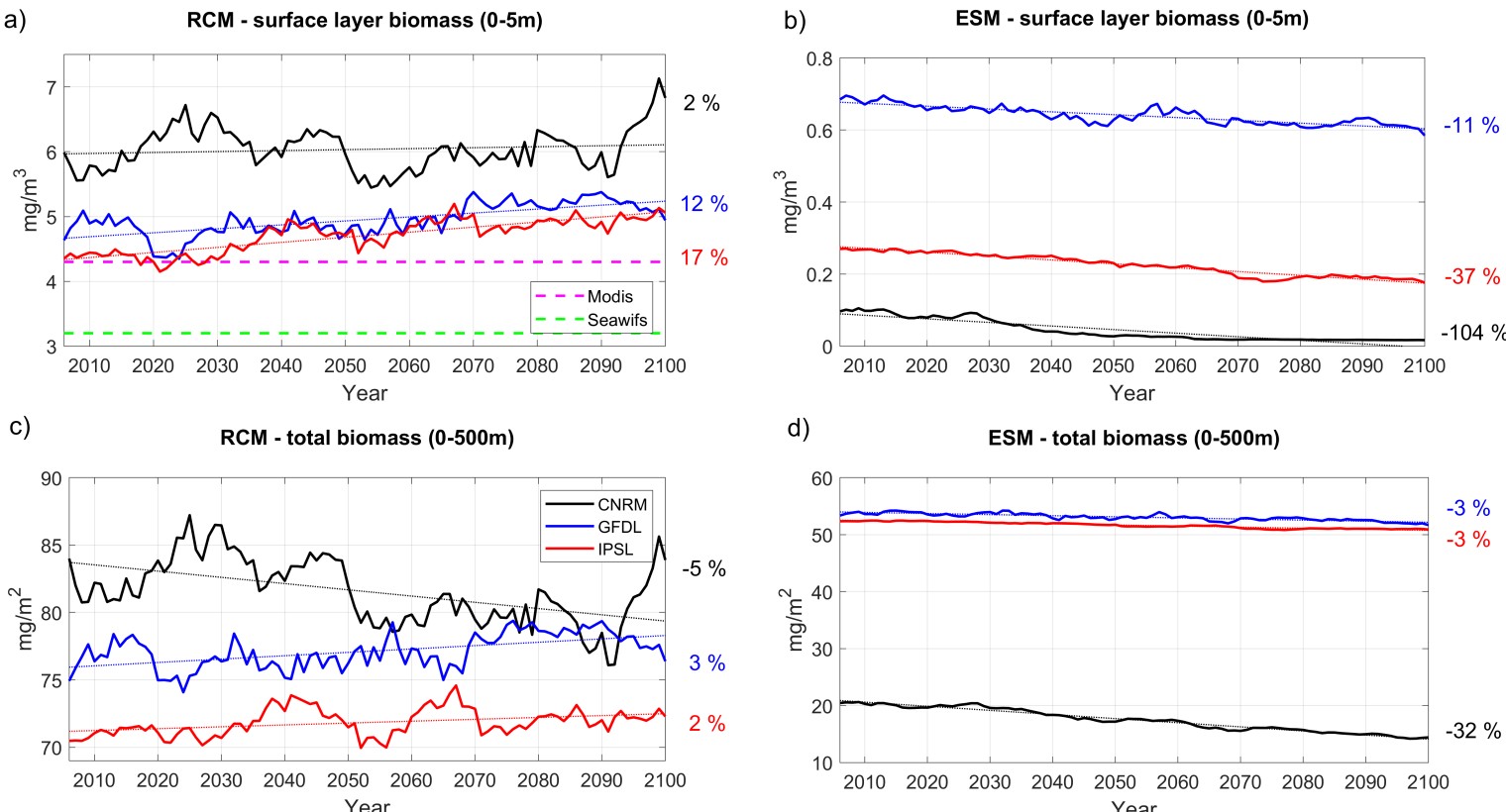

**Figure 12:** Surface chlorophyll (0-5m, in mgChl.m$^{-3}$) from (a) RCMs and (b) ESMs; depth-averaged chlorophyll concentration (0-50m, in mgChl.m$^{-2}$) from (c) RCMs and (d) ESMs. Color code: CNRM in black, GFDL in blue, IPSL in red. The timeseries are filtered using a 10-year moving average. Thin dotted colored lines indicate the linear trends. All variables are averaged in a coastal box (see Figure 2a). Dashed green and magenta lines in (a) mark the mean surface chlorophyll from SeaWIFs (1997-2010) and MODIS (2002-2015) respectively.

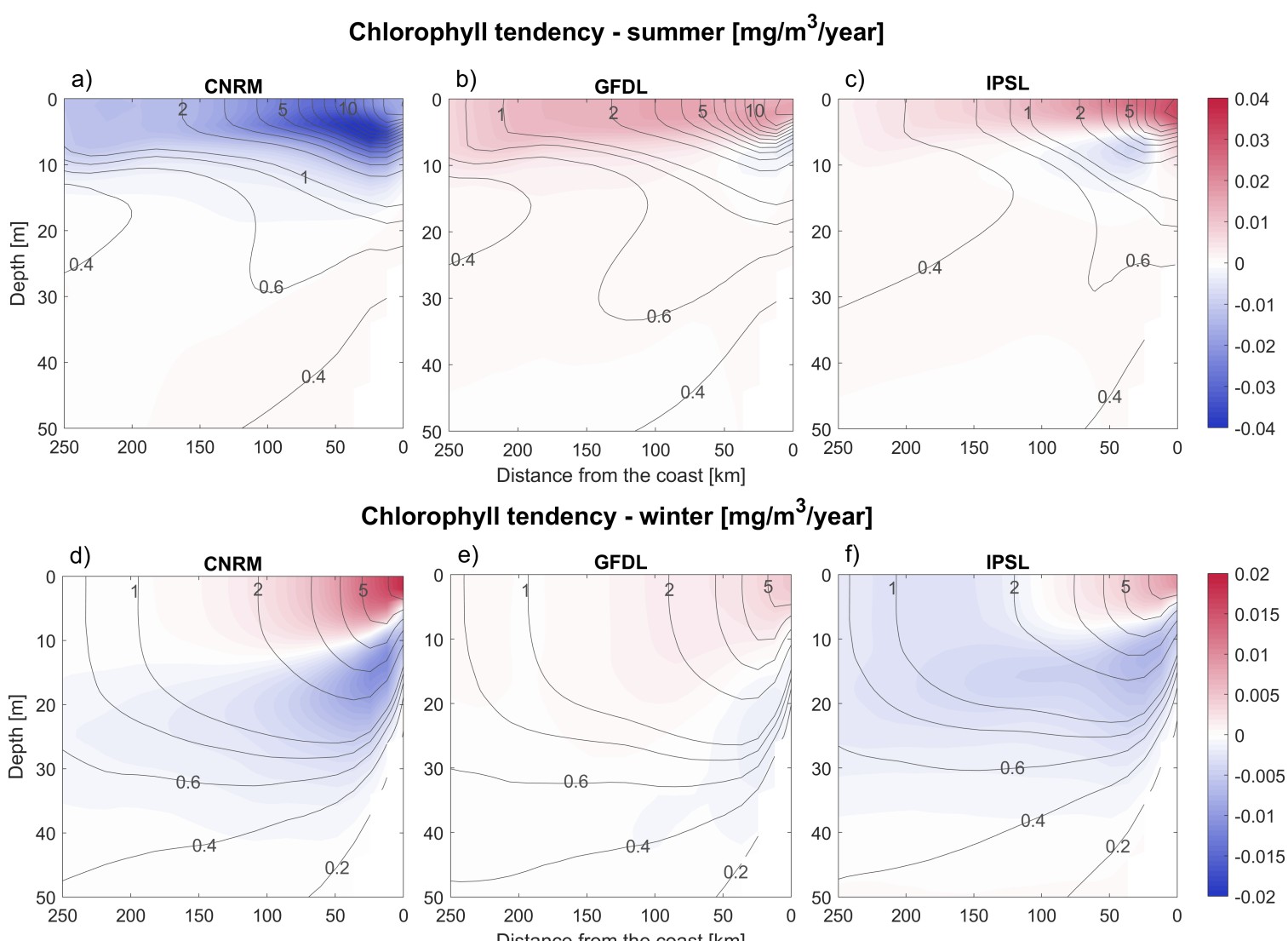

**Figure 13:** (a-c) Austral summer and (d-f) winter vertical sections of the RCM chlorophyll linear trends (in mg.m$^{-3}$ year$^{-1}$). The vertical cross-shore section corresponds to an alongshore average of cross-shore sections between 7°S and 13°S. Contours represent the mean control values (2006-2015).

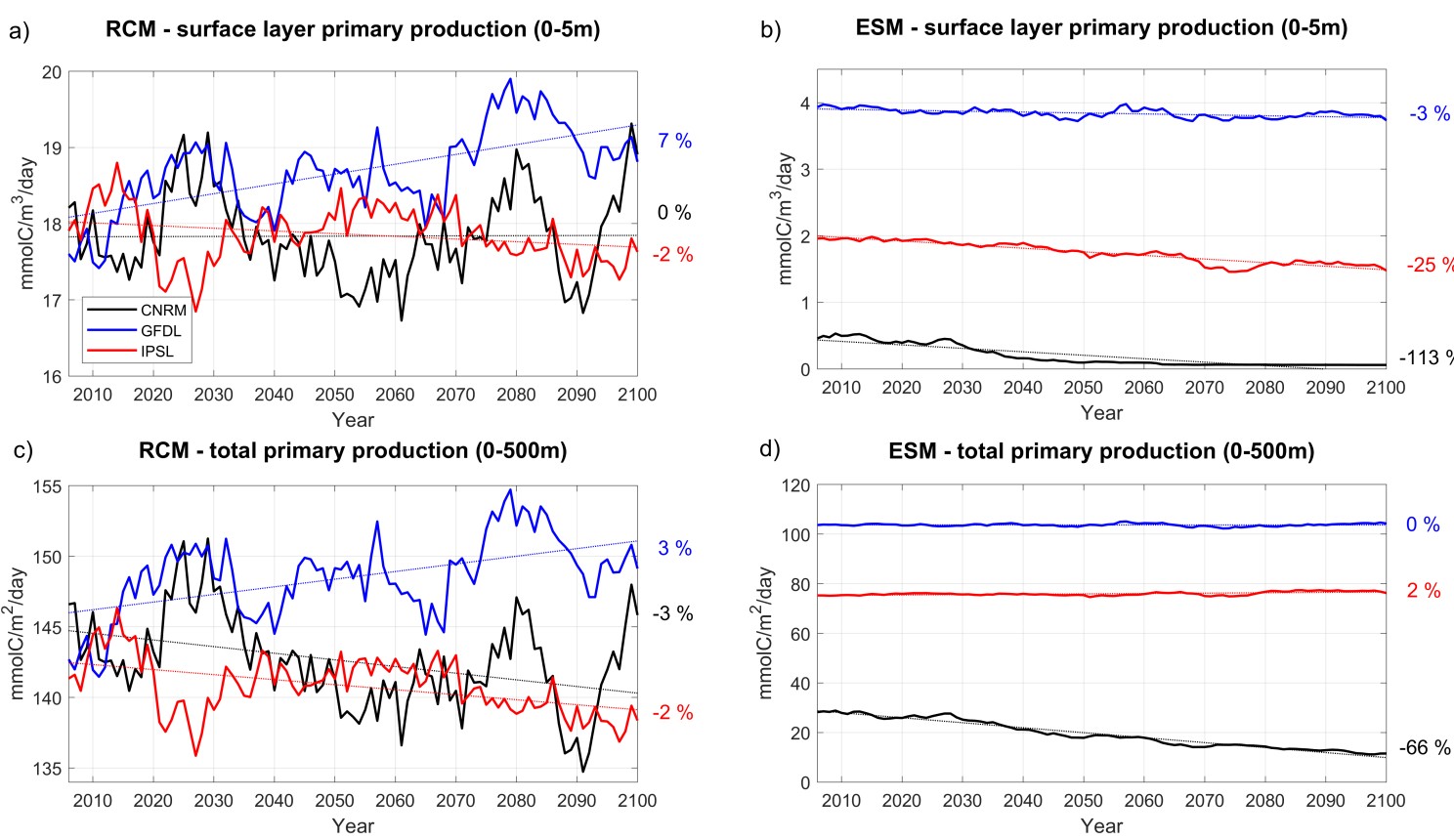

**Figure 14:** Same as 12 but for primary production (in mmolC.m$^{-3}$.day$^{-1}$) for the 0-5m surface layer in (a) and (b) and for the depth-integrated values (in mmolC.m$^{-2}$.day$-1$) in (c) and (d).

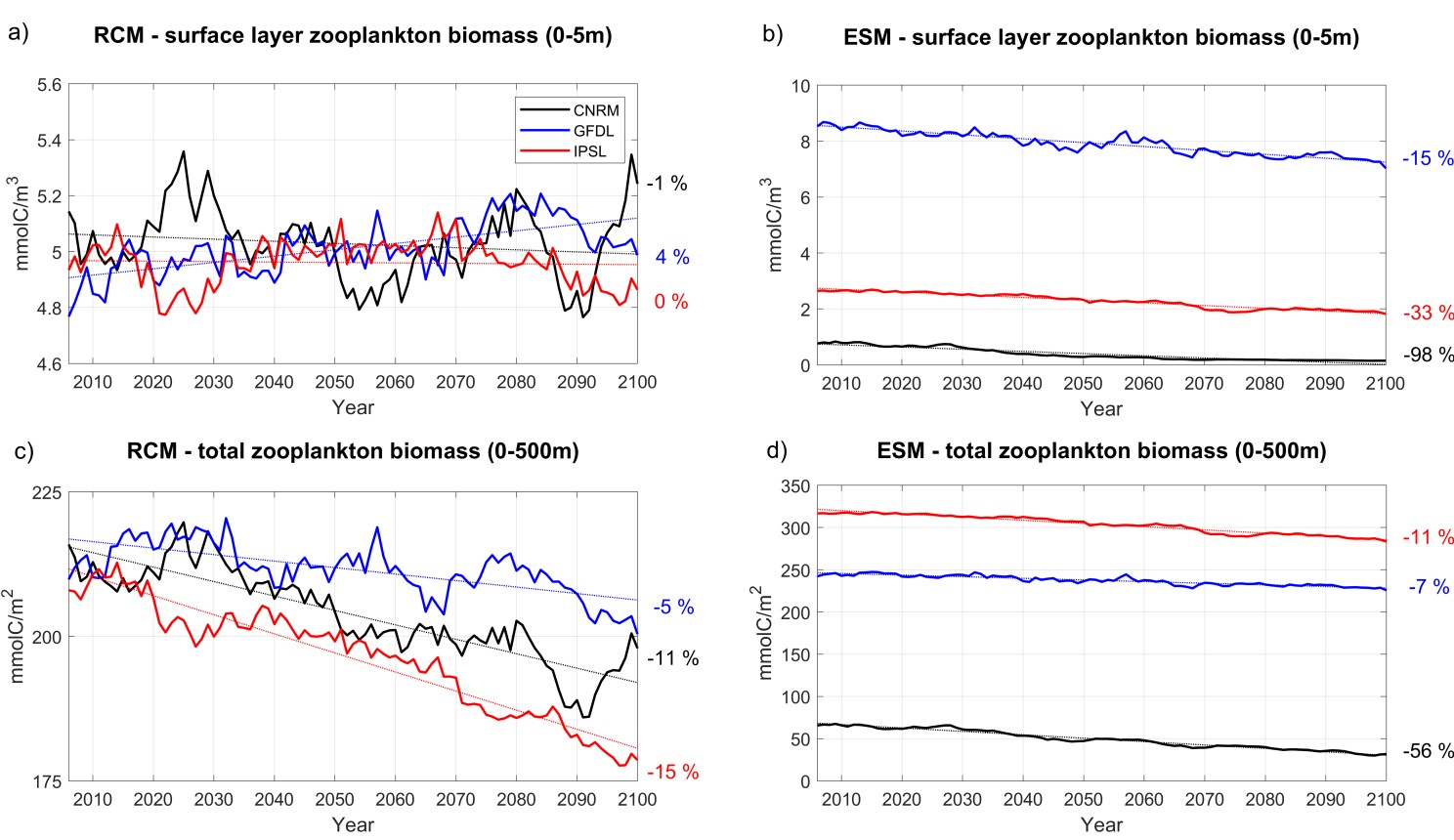

**Figure 15:** Same as 12 but for zooplankton (in mmolC.m$^{-3}$) for the 0-5m surface layer in (a) and (b) and for the depth-integrated values (in mmolC.m$^{-2}$) in (c) and (d).

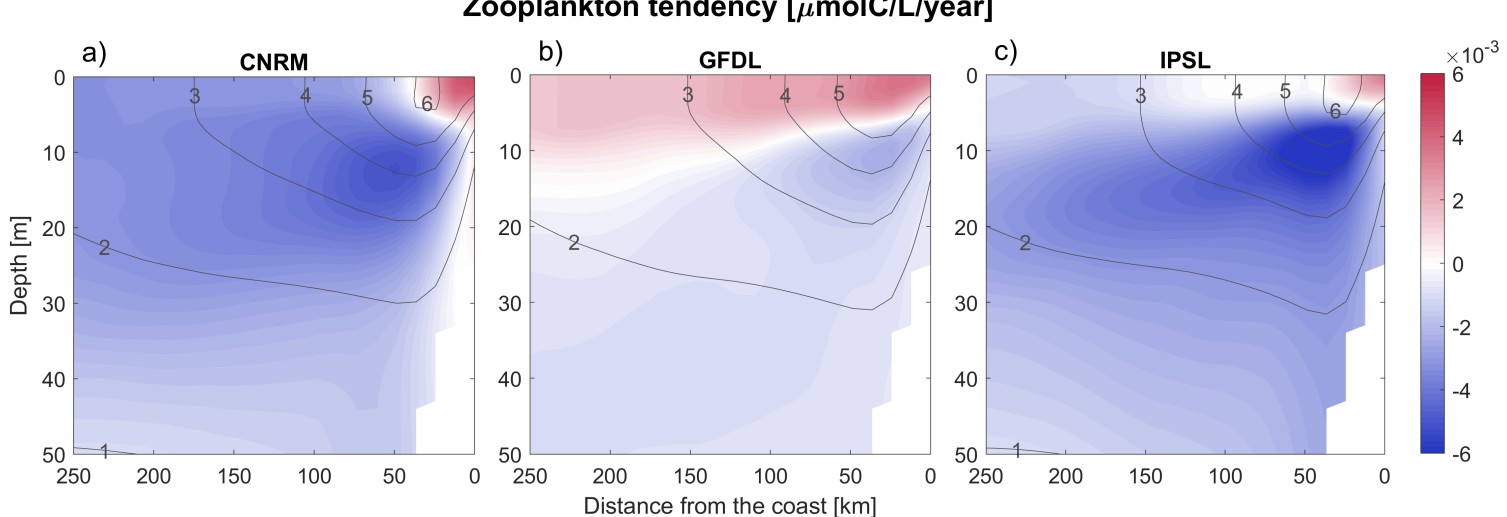

**Figure 16:** (a-c) Annual vertical sections of the RCM zooplankton linear trends (in $\mu molC.L^{-1}$ $year^{-1}$). The vertical cross-shore section corresponds to an alongshore average of cross-shore sections between $7°S$ and $13°S$. Contours represent the mean control values (2006-2015).

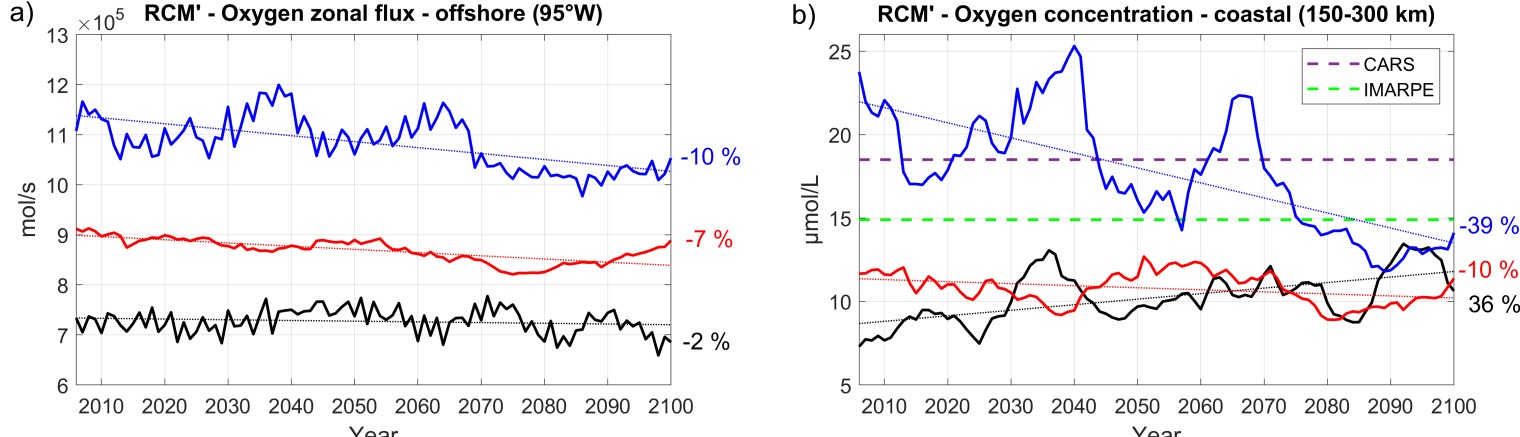

**Figure 17:** (a) Oxygen flux (in mol.L$^{-1}$, positive eastward) at 95°W, averaged between 2°N-10°S and 50m-200m and (b) nearshore subsurface oxygen content (averaged between 100 m and 200m in coastal box between 150 and 300 km from coast) for the three RCM' simulations (CNRM' in black, GFDL' in blue, IPSL' in red). RCM' simulations are forced by WOA climatological boundary conditions for oxygen. The timeseries are filtered using a 10-year moving average. CARS (dashed purple line) and IMARPE (dashed green line) climatological values are also shown

| ESM | Temperature | Salinity | Oxygen | Nitrate | Silicate | Phosphate |
|:---:|:---:|:---:|:---:|:---:|:---:|:---:|
| CESM1-BGC | 6% | 0.5% | 67% | 20% | 21% | 24% |
| IPSL-CM5A-LR | 13 % | 0.3% | 133% | 24% | 12% | 30% |
| IPSL-CM5B-LR | 5 % | 0.3% | 122% | 34% | 12% | 41% |
| **IPSL-CM5-MR** | **15%** | **0.5%** | **83%** | **20%** | **16%** | **28%** |
| **CNRM-CM5** | **7%** | **2.1%** | **150%** | **17%** | **27%** | **52%** |
| GFDL-ESM2G | 25% | 2.1% | 60% | 33% | 220% | 19% |
| **GFDL-ESM2M** | **8%** | **1.2%** | **47%** | **11%** | **61%** | **9%** |
| NOR-ESM1 | 13% | 1.9% | 60% | 46% | 88% | 36% |

**Table 1:** Normalized biases of the ESMs compared to WOA. They were calculated at each depth as $100 * \frac{|X_{model}(z) - X_{obs}(z)|}{X_{obs}(z)}$ and then averaged over 500 meters depth.

| Earth System Model | Ocean Model | Biogeochemical Model | Horizontal Resolution | Number of vertical levels | References |
|---|---|---|---|---|---|
| IPSL-CM5A-MR | NEMOv3.2 | PISCES (2 Pg, 2 Zg, N, P, Si, Fe, O2) | $\Delta x = 2°$ $\Delta y = 0.5\text{-}2°$ | 31 levels (10 m) | Dufresne et al. (2013) |
| CNRM-CM5 | NEMOv3.2 | PISCES (2 Pg, 2 Zg, N, P, Si, Fe, O2) | $\Delta x = 1°$ $\Delta y = 0.3\text{-}1°$ | 42 levels (10 m) | Voldoire et al. (2013) |
| GFDL-ESM2M | MOM4p1 | TOPAZ2 (3 Pg, 3 Zg, N, P, Si, Fe, O2) | $\Delta x = 1°$ $\Delta y = 0.3\text{-}1°$ | 50 levels (10 m) | Dunne et al. (2012) |

**Table 2:** Characteristics of the ESMs selected for the regional downscaling. The values in parenthesis indicate the thickness (in meters) of the surface layer in the ESM ocean model. Pg and Zg abbreviate "phytoplankton group" and "zooplankton group", respectively.

| | Wind stress | Shortwave radiation | Net longwave radiation | Downward longwave radiation | Mixed layer depth | Coastal SST (Kelvin) | ESM Coastal SST (Kelvin) |
|---|---|---|---|---|---|---|---|
| CNRM | **-10.9 ± 1.4%** R² = 0.65 | **-6.8 ± 0.3%** R² = 0.95 | **37.8 ± 1.2%** R² = 0.96 | **10.0 ± 0.4%** R² = 0.97 | **-18.5 ± 2.0%** R² = 0.73 | **1.20 ± 0.05%** R² = 0.95 | **1.18 ± 0.05%** R² = 0.95 |
| GFDL | **1.7 ± 0.8%** R² = 0.13 | **-4.4 ± 0.6%** R² = 0.66 | **28.4 ± 0.9%** R² = 0.97 | **6.7 ± 0.2%** R² = 0.98 | **-16.2 ± 1.1%** R² = 0.84 | **0.67 ± 0.03%** R² = 0.93 | **0.78 ± 0.02%** R² = 0.97 |
| IPSL | **-9.5 ± 1.2%** R² = 0.67 | **-0.5 ± 0.2%** R² = 0.18 | **21.1 ± 0.8%** R² = 0.96 | **9.8 ± 0.4%** R² = 0.98 | **-26.3 ± 1.3%** R² = 0.91 | **1.58 ± 0.05%** R² = 0.97 | **1.57 ± 0.04%** R² = 0.98 |

| | 20°C depth 95°W | 20°C depth coast | ESM 20°C depth coast | Velocity along x at 95°W | Offshore flux | Geostrophic flux | Ekman transport |
|---|---|---|---|---|---|---|---|
| CNRM | **25.7 ± 3.1%** R² = 0.77 | **101 ± 16%** R² = 0.77 | **101 ± 10%** R² = 0.89 | *-0.2 ± 2.5%* *R² = 0.00* | **-22.8 ± 2.8%** R² = 0.65 | *3.0 ± 4.2%* *R² = 0.02* | **-11.7 ± 1.2%** R² = 0.67 |
| GFDL | **5.2 ± 1.4%** R² = 0.27 | **11.5 ± 7.2%** R² = 0.09 | **20.9 ± 3.9%** R² = 0.53 | **-14.3 ± 1.6%** R² = 0.64 | *-0.5 ± 2.8%* *R² = 0.00* | *-3.0 ± 2.7%* *R² = 0.03* | **1.6 ± 0.8%** R² = 0.09 |
| IPSL | **70.6 ± 6.6%** R² = 0.91 | **207 ± 51%** R² = 0.88 | **126 ± 12%** R² = 0.93 | **-9.4 ± 1.3%** R² = 0.58 | **-25.1 ± 1.6%** R² = 0.85 | *0.3 ± 2.3%* *R² = 0.00* | **-11.0 ± 1.2%** R² = 0.70 |

**Table 3:** Differences (in %) between 2100 and 2006 (with respect to value in 2006) computed from RCM and ESM linear trends and R$^2$ for: wind stress, shortwave radiation, net longwave radiation, mixed layer depth, coastal SST for RCM and ESM, 20°C isotherm depth at coast for RCM and ESM and at 95°W for RCM, zonal velocity at 95°W, offshore and geostrophic fluxes. Bold font indicates significant values with a 90% level of confidence.

| Oxygen | | | | | |
|---|---|---|---|---|---|
| Coast 100-200 m | Coast 200-400 m | flux 95°W 50-200 m | Oxycline 22 µmol/L | Oxycline 44 µmol/L | Oxycline ESM |

| | Coast 100-200 m | Coast 200-400 m | flux 95°W 50-200 m | Oxycline 22 µmol/L | Oxycline 44 µmol/L | Oxycline ESM |
|---|---|---|---|---|---|---|
| CNRM | **483 ± 110%** R² = 0.96 | **722 ± 709%** R² = 0.92 | **32.8 ± 3.1%** R² = 0.80 | **209 ± 27%** R² = 0.93 | **119 ± 12%** R² = 0.91 | **85.0 ± 4.3%** R² = 0.96 |
| GFDL | *-4.1 ± 5.3% R² = 0.01* | **-14.3 ± 7.3%** R² = 0.06 | **-4.7 ± 2.0%** R² = 0.11 | **-5.5 ± 3.4%** R² = 0.05 | **-20.9 ± 3.4%** R² = 0.44 | **-9.1 ± 2.4%** R² = 0.26 |
| IPSL | **-17.8 ± 2.5%** R² = 0.41 | **-48.2 ± 2.0%** R² = 0.84 | **-3.7 ± 0.8%** R² = 0.38 | **-7.3 ± 1.8%** R² = 0.23 | **-4.5 ± 2.1%** R² = 0.09 | **13.2 ± 1.6%** R² = 0.67 |

| Nitrate | | | | | |
|---|---|---|---|---|---|

| | Coast 40-100 m | 95°W | flux 95°W | vertical flux coast euphotic layer | Nitracline | ESM nitracline |
|---|---|---|---|---|---|---|
| CNRM | **-13.9 ± 0.6%** R² = 0.94 | **-18.3 ± 0.3%** R² = 0.98 | **-26.6 ± 1.3%** R² = 0.91 | **-23.7 ± 1.1%** R² = 0.92 | **82 ± 13%** R² = 0.76 | **96.8 ± 4.5%** R² = 0.96 |
| GFDL | **-2.0 ± 0.5%** R² = 0.31 | **-5.1 ± 0.3%** R² = 0.84 | **-17.8 ± 1.6%** R² = 0.69 | *-0.5 ± 1.2% R² = 0.14* | **-25.4 ± 3.6%** R² = 0.51 | *1.6 ± 2.5% R² = 0.01* |
| IPSL | **-10.4 ± 0.3%** R² = 0.96 | **-7.0 ± 0.3%** R² = 0.92 | **-20.3 ± 1.1%** R² = 0.91 | **-39.2 ± 1.9%** R² = 0.89 | **31.6 ± 2.9%** R² = 0.78 | **34.2 ± 2.2%** R² = 0.90 |

**Table 4:** Differences (in %) between 2100 and 2006 (with respect to value in 2006) computed from RCM and ESM linear trends and $R^2$ for: nearshore oxygen content, oxygen eastward flux, oxycline depth for RCM and ESM, nitrate content nearshore and at 95°W, nitrate eastward flux, upward nitrate flux into the euphotic layer and nitracline depth for RCM and ESM. Bold font indicates significant values with a 90% level of confidence.

| | Surface Chlorophyll | | Total Chlorophyll | | Surface Zooplankton | | Total Zooplankton | |
|---|---|---|---|---|---|---|---|---|
| | RCM | ESM | RCM | ESM | RCM | ESM | RCM | ESM |
| CNRM | *2.3 ± 3.6%* *R² = 0.02* | **-104 ± 4.9%** R² = 0.83 | **-5.2 ± 1.6%** R² = 0.28 | **-32.5 ± 0.8%** R² = 0.96 | *-1.4 ± 1.6%* *R² = 0.03* | **-97.8 ± 3.8%** R² = 0.88 | **-10.9 ± 1.1%** R² = 0.78 | **-55.5 ± 1.1%** R² = 0.96 |
| GFDL | **12.3 ± 2.6%** R² = 0.48 | **-10.9 ± 0.9%** R² = 0.73 | **3.1 ± 0.9%** R² = 0.29 | **-3.4 ± 0.3%** R² = 0.72 | **4.3 ± 0.9%** R² = 0.44 | **-15.3 ± 0.7%** R² = 0.86 | **-4.9 ± 1.0%** R² = 0.48 | **-7.4 ± 0.4%** R² = 0.85 |
| IPSL | **17.0 ± 1.8%** R² = 0.71 | **-36.6 ± 1.4%** R² = 0.94 | **1.8 ± 0.5%** R² = 0.15 | **-3.4 ± 0.2%** R² = 0.93 | *-0.3 ± 1.0%* *R² = 0.00* | **-33.0 ± 1.2%** R² = 0.94 | **-14.7 ± 0.7%** R² = 0.92 | **-11.1 ± 0.4%** R² = 0.95 |

**Table 5:** Differences (in %) between 2100 and 2006 (with respect to value in 2006) computed from RCM and ESM linear trends and $R^2$ for chlorophyll and zooplankton. Total chlorophyll and total zooplankton indicate depth-integrated values over 0-500m. Bold font indicates significant values with a 90% level of confidence.

# Supplementary

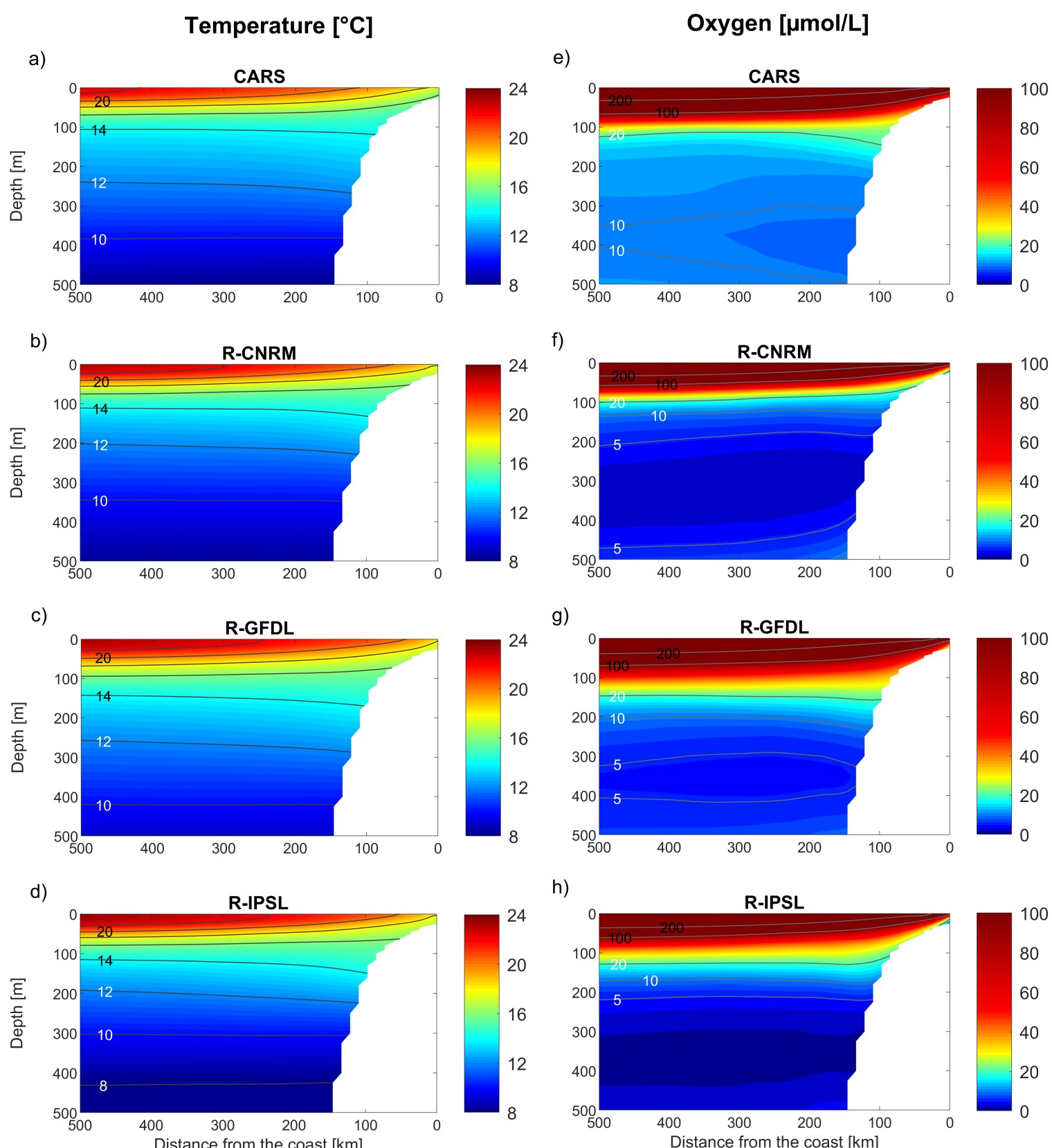

**Figure S1:** Annual vertical sections of (a),(b),(c),(d) nearshore temperature and (e),(f),(g),(h) nearshore oxygen for (a),(e) CARS, (b),(f) R-CNRM, (c),(g) R-GFDL and (d),(h) R-IPSL. They are alongshore averaged between 7°S and 13°S, and from 2006 to 2015.

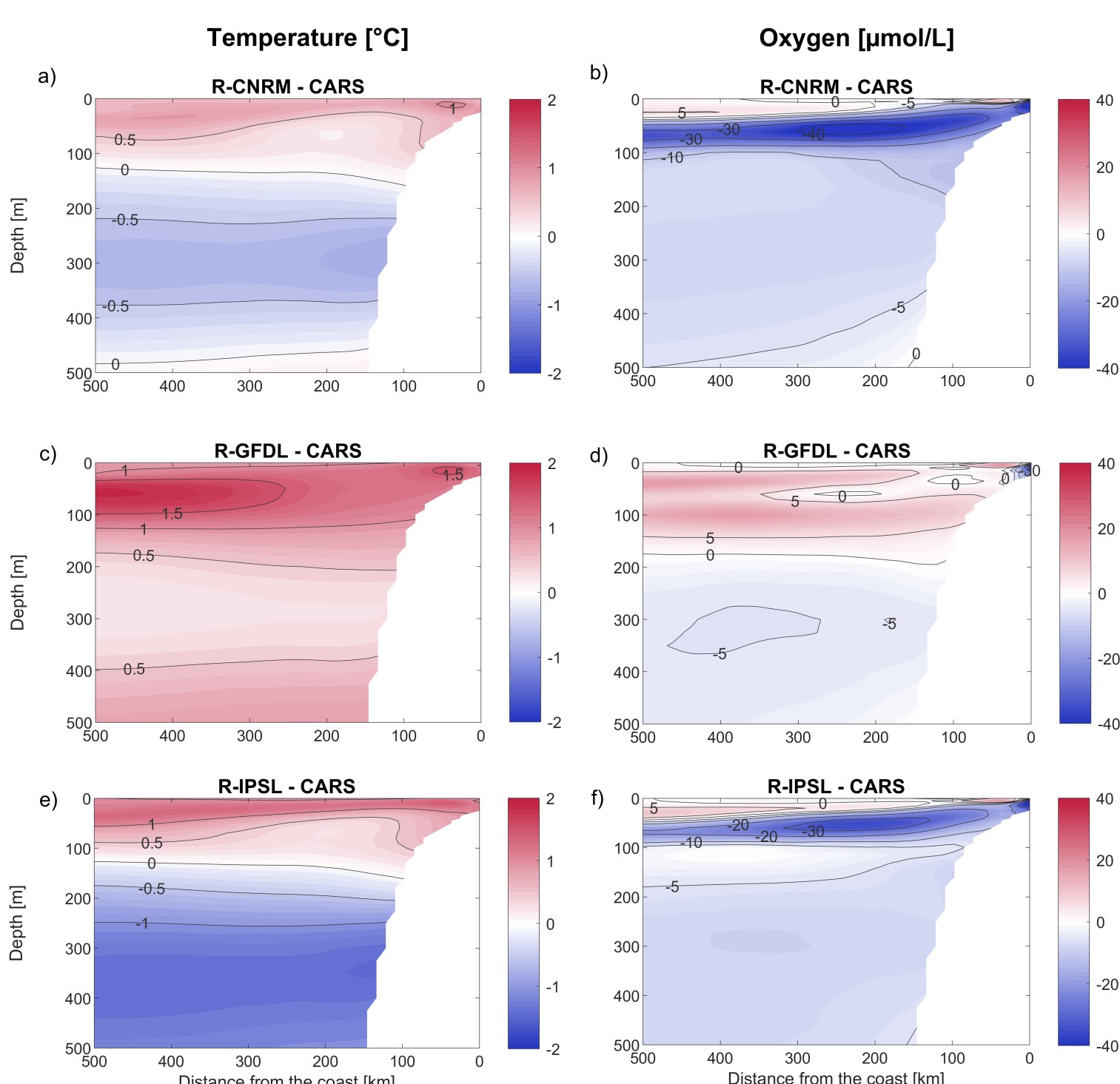

**Figure S2:** Annual vertical sections of bias of (a),(b),(c),(d) nearshore temperature and (e),(f),(g),(h) nearshore oxygen. The differences are calculated between (a),(d) R-CNRM and CARS, (b),(e) R-GFDL and CARS and (c),(f) R-IPSL and CARS. They are alongshore averaged between 7°S and 13°S, and from 2006 to 2015.