# Peer review of "Physical and biogeochemical impacts of RCP8.5 scenario in the Peru upwelling system"

_Biogeosciences, 2020_

## Referee Comment (RC1) · Anonymous Referee #1 · 18 Feb 2020

**General comments**

In this manuscript the authors analyze changes in the ocean physics and biogeochemistry of the Peru upwelling system using regional projections to the end of the century. They use downscaled output from three different earth system models to initialize and force one regional, high-resolution ocean model under the RCP 8.5 "business as usual" scenario.

Regional downscaling is an accepted methodology for assessing future impacts in regions like the upwelling region off Peru, where global scale models are unable to accurately resolve the process of upwelling. Additionally, model uncertainty in global projections can be substantial, and the use of three downscaled global models can give a clearer measure of such uncertainty in coastal regions, even though only one

**C1**

**regional model was employed.**

The work and analysis presented here can be a solid contribution to regional climate change literature and is worth publishing. However, I think that some work needs to be done in order to provide the reader with a measure of uncertainty on the model performance and the projected changes by the end of the century. Because I think that this is so necessary, I recommend major revisions before publishing in biogeosciences. Here my two main concerns:

1) I think that a more quantitative and critical evaluation of the regional model performance after the spinup is missing. A clear discussion of how well the "baseline", or present day is being represented in the regional simulation is necessary, in particular for sensitive parameters such as thermocline and oxycline depth.

2) Some measure of uncertainty in the percentage of change by the end of the century for each variable is needed. This percentage values form most of the base of the whole discussion and are calculated on the basis of linear trends. A quantitative estimate of how well a linear model fits the timeseries examined, or perhaps an estimate of the actual temporal variability around the trend could make the interpretation of the long-term changes more robust.

**Specific comments**

Section 2.3: To choose the global model for regional downscaling, the authors use averaged vertical profiles of a meridional section and compare the bias with an observation-based gridded product (World Ocean Atlas 2009). It is not clear to me if the model was sampled to represent the time period of WOA09, which years is the WOA09 climatology representing?

Some ideas in section 2.3 need to be more quantitative. E.g. phrases like "too low", "realistic enough" are somewhat subjective. The authors mention that the temperature and salinity biases are weak, but what does weak mean? How do we compare the

weak salinity and temperature biases to the biogeochemical biases?

Lines 293-298: The authors describe a shoaling of the mixed layer depth in all simulations and the agreement or disagreement with a gridded product. I find this confusing since this idea comes after they mention that the "thickness of the surface layer more than doubles" (line 287). Also, they note that the mixed layer is calculated differently in the model and in the gridded product. How is the mixed layer calculated in the model then?

Line 279: The term thermocline depth needs to be clearly defined as the isotherm of 20C, as is indicated in figure 6 and as was done with the oxycline (line 341) or nitracline.

Lines 333-339: In the text, they mention that figure 10 shows the evolution of nearshore DO concentration, but the trends in this figure are calculated over a region that differs from the coastal box used through the analysis. There is no mention or explanation of why these trends were calculated in an oceanic box that differs in size and distance from the coast than the rest of the analysis.

Line 382: Positive trends in surface biomass were found in R-GFDL and R-IPSL, but the nitracline only deepens in R-IPSL, in R-GFDL the nitracline gets shallower. The increase in surface biomass would be surprising only in R-IPSL.

Typos and minor issues

Line 21: The resolution of the model is not consistent through the text, In the abstract is 10 km, but in the description of the model (line 100) is  $\sim$  12 km.

Line 31: "small pelagic fisheries"

Line 50: IPCC is not defined

Line 52: "Oyarzún"

Line 58: AR is not defined

СЗ

Line 76: change 2017 for 2018.

Lines 82-83: The phrase "most recent climate scenarios" is not clear to me. Do you imply that the RCP's are recently developed scenarios? that we are following these scenarios? please clarify.

Lines 117-118: Is it possible to fix the exponential with the symbol and superscript?

Line 124: CMIP5 is not defined

Line 141: Needs a comma after "However"

Line 171: There is no entry on the reference list for Echevin et al., 2010.

Line 200: Section 2.7 is missing

Line 216: The number of the figures they are referring to is missing.

Lines 255-262: This section is described as if the trends where those of the ESMs, when figure 4 shows the change in the RCMs. Also, there is no consistency with the use of "R+model" to indicate the downscaled simulation.

Line 316: Another example of a subjective phrase "weak dissolved O2 concentrations".

Line 318: There is no entry for Espinoza et al., 2019 in the reference list.

Line 321: You mean the RCM eastward surface flow?

Line 328: "The trend is relatively weak..."

Line 330: I find that the use of parentheses to indicate the opposite of an idea in a paragraph is confusing and inefficient. I invite the authors to use parentheses for clarification and citations only and not to save space. See Robock, A. 2010. Parentheses are (are not) for references and clarification (savings space). Eos, Trans. Amer. Geophys. Union, 91(45): 419).

Table 1. Needs a better description of terms. What does 10 m mean? 10 m wind?

Fig. 1. For clarity, I would suggest to make the vertical axis of each subplot equal and visualizing the extent of the influence of the OMZ on nitrate is not evident. Also, the thickness of the lines representing the selected ESM's is not really different from the rest. Perhaps the legend should refer to these as "solid colored lines" instead of "thick colored lines."

Fig. 2. The description of the legend is not consistent with what is being showed and what is described on the text. i.e., b) and d) should be output from the RCM (downscaled).

Fig. 3. The word "value" is missing in the legend just before (c).

Fig. 4. In the legend (c) is missing.

Fig. 11. In the legend, fix the superscript in  $\mu$ mol L-1.

Fig. 16. The legend is wrong, there are no figures 16d-f.

Fig. 17. In a) the title of the figure is wrong. These should be the trends of the ESMs not RCM as mentioned in the legend and in the text (line 508). It should be indicated somewhere in the legend that the trends in b) and c) correspond to the R-GCM' sensitivity experiments.

---

## Referee Comment (RC2) · Michael Jacox (Referee) · 31 Mar 2020

General Comments

The authors explore the projected physical and biogeochemical state of the Northern Humboldt Current System (NHCS) under future climate change using a regional circulation model (RCM) forced by three global earth system models (ESMs). They describe changes in a range of ocean properties from temperature to zooplankton biomass, focusing on trends relative to historical conditions as well as the differences among the different ESM and RCM projections.

Future conditions in eastern boundary upwelling systems like the NHCS are of considerable interest due to the biogeochemical, ecological, and socioeconomic importance

of these regions. It's also well known that fine scale dynamics in these regions are important and are not well captured by coarse resolution global models, so there is interest in the potential added value provided by dynamical downscaling. Therefore, this is valuable work and is at the cutting edge of regional ocean projection. The inclusion of biogeochemistry, the use of multiple ESMs to force the regional model, and the bias correction of the forcing are all notable and positive elements of the research.

The manuscript is mostly descriptive; the authors note that further mechanistic analysis is left to future research. In my view, the most important results are the comparisons of projected changes between the global and regional models. We know global models have biases, but it's when the projected change is altered by downscaling that a stronger case is made for the need to downscale. The authors find that this is the case for biogeochemical, but not physical, variables. I have a number of specific comments below, but my main concerns are with several choices in the methods, detailed below.

Specific Comments

I have three main concerns on the methods:

1. (Section 2.3). The choice of which ESMs to use has been justified based on historical comparisons with observations. However, there is a growing body of research arguing against this method, since these historical model evaluations do not necessarily correspond to how well a model captures the response to future climate forcing. "Emergent constraints" have been offered as a more relevant method for evaluating climate models (Hall et al. 2019). In the absence compelling reasons why a model is unrealistic for the future change, the default should be to pick a suite of models that capture the range of potential futures.

2. (Section 2.4). As I understand it, this method produces forcing with no high-resolution (sub monthly) variability. High frequency wind variability can be very important especially to the BGC in Eastern Boundary systems. For example, Gruber et al (2006) attribute model chlorophyll biases to the use of monthly forcing. For future

projections, one can add representative high frequency variability (e.g., from historical reanalysis) as a third term on the right hand side of equation (1). Similar has been done for historical sensitivity analyses (Frischknecht et al. 2015, Jacox et al. 2015).

3. (Section 2.5): First, it's unclear why one would not bias-correct the physical ocean boundary conditions. For consistency they should be treated like the surface and ocean BGC fields. Second, oxygen should be treated the same as the other biogeochemical variables. While I understand the concern about unrealistic oxygen values, the oxygenation trend is inextricably linked to the trend in nitrate concentration (Fig. 11) and presumably other nutrients, and in turn with trends in productivity. It doesn't make sense to deem the oxygen trend unrealistic and the others realistic. Furthermore, since oxygen and nitrate variability are closely coupled, imposing the ESM change in one but not the other introduces biogeochemical inconsistencies that may compromise the RCM findings. The analysis of oxygen using climatological boundary conditions is still interesting as it allows one to separate different contributions to the regional change, but it's not consistent with the rest of the analysis. Therefore, the main text should include the GCM change, with the context that you are trying to bound the range of possible futures, not to predict exactly what happens in the future. The oxygen analysis using climatological boundary conditions can move to discussion.

Detailed Comments:

L20, 428, 547: Suggest removing "business as usual": See Hausfather and Peters (2020).

L87: Unclear what "in the following paragraphs" refers to. The whole rest of the paper?

L115-121: The differences are described and are stated to be important, but it's not clear what is the motivation for these changes.

Section 2.6: The temporal coverage of these data sets is quite short for evaluating historical model performance, given that the decadal variability in the ESMs should not

be expected to align with nature. Something like 30 years would be more appropriate, but in any case the authors should be wary of caveats associated with using short observational records.

L205: This is probably fine as a proxy, but it's worth noting that it doesn't explicitly represent upwelling, including the curl-driven component. If so inclined, one could get a more accurate upwelling metric by integrating the Ekman and geostrophic components over the region of interest or by using the vertical velocity at the based of the Ekman layer (Jacox et al. 2018). It would also be helpful here to describe the calculation of the cross-shore geostrophic transport.

L221: Thanks to its high spatial resolution and the bias correction of the forcing.

Fig. 3: I think it would be more appropriate to show the bias corrected ESM change (i.e., remove the mean ESM SST bias so that they all start from the same place). I also don't think a % change is best for SST, at least if the units are Celsius. In Fig. 3 the ESM % changes are lower because they are starting from a warm-biased state. But the magnitudes of projected temperature changes are as large as or larger than the RCM. Lastly, throughout the manuscript some indication of significance should be added to the trends.

L248-249: Did Bakun actually project cooling, or just intensified upwelling? There could be intensified upwelling but still warming due to dominance of the surface heating.

L252-253: It's not clear to me the evidence that this pattern results from the upwelling and subsequent lateral transport/damping of subsurface anomalies.

Fig. 4: Would be informative to see the net longwave and shortwave radiation (not just downwelling).

L266: Initially it seems strange that the offshore transport trend is 2x greater than the wind stress trend, since the transport is linearly related to the wind stress. But, it does make sense because when you calculate the Ekman transport (i.e., Fig. 5a minus Fig.

5b), the change is ~10%, consistent with the winds. This should be explained in the text, and I suggest adding the Ekman transport as a third panel to Fig. 5.

L269-270: It's also interesting that since there's a long-term trend in Ekman transport but not in geostrophic transport, the relative contribution of the geostrophic transport increases over time

L275: Do you mean they are locally influenced by the passage of waves? Or are you suggesting the waves actually propagate (advect) the anomalies somehow?

L279: I would be careful about equating the depth of an isotherm (D20) with the depth of the thermocline (i.e., the depth of maximum temperature gradient). Temperature biases (or changes) will alter D20 but not necessarily the thermocline depth.

L297: How is MLD calculated?

Figure 8: I would like to see the ESM tendencies on here as well

L308-309: I don't understand why this statement is here. I would delete it

L351 and elsewhere: "deemed realistic enough" isn't very convincing. I don't think you have to argue for the realism of the ESM changes, rather you are looking at the regional impact of the ESM changes as one potential future scenario.

L352: Since the 95W location is discussed a number of times, it would be helpful to show it on a map (e.g., Fig. 2a along with the coastal region)

L360-365: It's hard to compare a concentration is one place (Fig. 11a) with a depth level in another place (Fig. 11c), especially when arguing that one is the driver of the other. Can these be presented in a more consistent way?

L389-397: I must say I'm surprised to see trends of opposite sign in the upper 10m. Surely one can't have opposite trends at different depths within the mixed layer. Perhaps this is a seasonal signature, e.g., in summer the mixed layer is very shallow (~5m) and increased chlorophyll in the seasonal mixed layer is driving the overall trend. But

the authors should look at this in more detail to explain how chlorophyll at 2m can have an opposite trend from chlorophyll at 7m.

L430: This may be true, but without a heat budget it's speculative. There will be other contributions as well (e.g., local surface fluxes). The text at L456-460 is good.

L547-552: There is also a summary statement like this in the previous section (L427-429). One of them should be cut – probably the earlier one.

L561-565: I think this is all speculation, so should be presented as hypotheses rather than fact (unless there is evidence to support it)

Technical Corrections

L117-118: Does the period in a.T indicate multiplication?

L368-369: Quasi-absent doesn't make sense. Maybe negligible? Insignificant?

Table 1: Suggest including in the caption the meaning of abbreviations (mainly Pg and Zg) and the meaning of (10m) in the number of vertical levels column. Also I don't think the full references are needed in the table, they can be in the reference list.

Figure 4: in caption (d) should be (c)

Figure 6, 7, 13, 16: Values should be positive for depth

Figure 17: Top panel should be ESM?

References

Frischknecht, M., M. Münnich, and N. Gruber (2015), Remote versus local influence of ENSO on the California Current System, J. Geophys. Res. Oceans, 120, 1353–1374, doi:10.1002/2014JC010531.

Gruber, N., H. Frenzel, S. C. Doney, P. Marchesiello, J. C. McWilliams, J. R. Moisan, J. J. Oram, G.-K. Plattner, and K. D. Stolzenbach (2006), Eddy-resolving simulation of plankton ecosystem dynamics in the California Current System, Deep Sea Res., Part

I, 53, 1483–1516.

Hall, A., Cox, P., Huntingford, C., & Klein, S. (2019). Progressing emergent constraints on future climate change. Nature Climate Change, 9(4), 269-278.

Hausfather, Z., & Peters, G. P. (2020). Emissions–the 'business as usual' story is misleading. https://www.nature.com/articles/d41586-020-00177-3

Jacox, M. G., S. J. Bograd, E. L. Hazen, and J. Fiechter (2015), Sensitivity of the California Current nutrient supply to wind, heat, and remote ocean forcing, Geophys. Res. Lett., 42, doi:10.1002/ 2015GL065147.

Jacox, M. G., Edwards, C. A., Hazen, E. L., & Bograd, S. J. (2018). Coastal upwelling revisited: Ekman, Bakun, and improved upwelling indices for the US West Coast. Journal of Geophysical Research: Oceans, 123(10), 7332-7350.

---

## Author Comment (AC1) · 5 May 2020

We thank the reviewer for his/her constructive comments. Due to a comment from Reviewer#2 arguing that the results from the regional simulations forced by ESM oxygen trends (labelled RCM' in the previous version of the manuscript) should be presented before the simulations forced by climatological oxygen boundary conditions, the paper has been thoroughly reorganized. The figures and tables presenting the biogeochemical trends have been modified. This does not change the general message of our paper, but large portions of the text have been modified.

Comments:

C1) I think that a more quantitative and critical evaluation of the regional model perfor

mance after the spinup is missing. A clear discussion of how well the "baseline", or present day is being represented in the regional simulation is necessary, in particular for sensitive parameters such as thermocline and oxycline depth.

R: We have added new figures showing cross-shore sections of mean state and bias of temperature and dissolved oxygen (DO) for the three regional simulations (period 2006-2015). The model is compared to CARS climatology (interpolated on the model's grid). These figures, included in the supplementary material, are briefly described in the text (lines 254-256 for temperature and lines 366-371 for DO)

C2) Some measure of uncertainty in the percentage of change by the end of the century for each variable is needed. This percentage values form most of the base of the whole discussion and are calculated on the basis of linear trends. A quantitative estimate of how well a linear model fits the timeseries examined, or perhaps an estimate of the actual temporal variability around the trend could make the interpretation of the long- term changes more robust.

R: We have estimated the trend uncertainty based on a bootstrap method. We construct 10 000 synthetic time series by randomly removing data points in the annual series. We converted the trend uncertainty into a percentage uncertainty, now reported in Tables 3,4,5. We also have computed the $R^2$ from the least square estimation in the tables. Most of the trends are significant at the 10% level. The significant trends are reported in bold font in the tables. We now explain how the uncertainty is computed in the methodology section 2.8.

Specific comments

C: Section 2.3: To choose the global model for regional downscaling, the authors use averaged vertical profiles of a meridional section and compare the bias with an observation-based gridded product (World Ocean Atlas 2009). It is not clear to me if the model was sampled to represent the time period of WOA09, which years is the WOA09 climatology representing?

R: The temperature and salinity from the CMIP5 historical simulations were averaged between 1950 and 2005 to compare with the WOA2009 climatology, which includes observations mainly collected between 1950s and 2009. The nutrient and oxygen profiles from the CMIP5 historical simulations were averaged between 1980 and 2005. They are compared to the WOA2009 which includes biogeochemical observations mostly in recent decades (i.e. after 1980) in the equatorial pacific.

C: Some ideas in section 2.3 need to be more quantitative. E.g. phrases like "too low", "realistic enough" are somewhat subjective. The authors mention that the temperature and salinity biases are weak, but what does weak mean? How do we compare the weak salinity and temperature biases to the biogeochemical biases?

R: The reviewer is right. First we added temperature and salinity profiles in Fig.1 to allow for visual comparison between the ESMs. Second, we computed a normalized bias, defined as: $NB(z)= |X_{model}(z)-X_{obs}(z)|/X_{obs}(z) \times 100$ for each variable $X(=T,S, \text{nutrients}, O_2)$. This allows to quantify the amplitude of the normalized bias between the ESMs and compare the normalized bias of different variables. The depth-averaged values of the normalized bias are reported in Table 1. We find that the normalized bias for temperature and salinity are weaker than those for nutrients and oxygen. We corrected the text to avoid vague terms and be more quantitative (see section 2.3).

C: Lines 293-298: The authors describe a shoaling of the mixed layer depth in all simulations and the agreement or disagreement with a gridded product. I find this confusing since this idea comes after they mention that the "thickness of the surface layer more than doubles" (line 287).

R: By surface layer we did not mean the mixed layer in this paragraph, but the surface layer with waters warmer than 20°C. We defined D20 in lines 303-304 and rephrased the sentence (line 312)

C: Also, they note that the mixed layer is calculated differently in the model and in the

gridded product. How is the mixed layer calculated in the model then?

R: The model surface boundary layer is computed from the value a critical Richardson number computed using the KPP formulation, whereas the observed mixed layer depth was computed from individual temperature profiles. However previous modelling work show that the surface boundary layer thickness is very close to the model mixed layer (Liu and Fox-Kemper, 2017). We added this information and this reference (lines 319-321)

C: Line 279: The term thermocline depth needs to be clearly defined as the isotherm of 20C, as is indicated in figure 6 and as was done with the oxycline (line 341) or nitracline.

R: We agree with the reviewer that this is unclear. As noticed by another reviewer, D20 and thermocline may be located at different depths. We now no longer refer to the thermocline, simply D20.

C: Lines 333-339: In the text, they mention that figure 10 shows the evolution of nearshore DO concentration, but the trends in this figure are calculated over a region that differs from the coastal box used through the analysis. There is no mention or explanation of why these trends were calculated in an oceanic box that differs in size and distance from the coast than the rest of the analysis.

R: We agree with the reviewer that some clarification is needed here. In this section we compare the nearshore DO content in the RCMs and ESMs between 100 and 200m depth. However, the coarse resolution and topography of the ESM implies that few grid points are present in this depth range in the 100 km band (in particular in GFDL). We believe that the comparison is thus more accurate in the 150km-300 km offshore band. In the same way as the oxycline is quite deep in R-GFDL we had to extend the width of the box to 200 km. This was not the case for the nitracline (depth of nitrate isosurface 21 umol) which was shallower and could thus be computed in the 0-100 km band (Fig.11d).We added explanation in lines 366-367.

C: Line 382: Positive trends in surface biomass were found in R-GFDL and R-IPSL, but the nitracline only deepens in R-IPSL, in R-GFDL the nitracline gets shallower. The increase in surface biomass would be surprising only in R-IPSL.

R: The reviewer is right. We corrected the text accordingly.

Typos and minor issues

Line 21: The resolution of the model is not consistent through the text, In the abstract is 10 km, but in the description of the model (line 100) is âĹij 12 km.

R: The resolution is 12 km. We corrected the error in the abstract.

Line 31: "small pelagic fisheries"

R: Corrected.

Line 50: IPCC is not defined

R: We replaced by CMIP5 and wrote out the meaning of the acronym line 51.

Line 52: "Oyarzún"

R: Corrected.

Line 58: AR is not defined

R: Corrected.

Line 76: change 2017 for 2018.

R: Corrected.

Lines 82-83: The phrase "most recent climate scenarios" is not clear to me. Do you imply that the RCP's are recently developed scenarios? that we are following these scenarios? please clarify.

R: We modified the sentence: "..under climate scenarios taking into account economic

and population growth assumptions (e.g. RCP8.5) and over longer time periods (e.g. 100 years)." (lines 85-86)

Lines 117-118: Is it possible to fix the exponential with the symbol and superscript?

R: Corrected.

Line 124: CMIP5 is not defined

R: It is now defined line 51.

Line 141: Needs a comma after "However"

R: Corrected.

Line 171: There is no entry on the reference list for Echevin et al., 2010.

R: Thank you for noticing this error, we added the correct reference (Echevin et al. 2012).

Line 200: Section 2.7 is missing

R: Corrected.

Line 216: The number of the figures they are referring to is missing.

R: Corrected.

Lines 255-262: This section is described as if the trends where those of the ESMs, when figure 4 shows the change in the RCMs. Also, there is no consistency with the use of "R+model" to indicate the downscaled simulation.

R: We corrected the text and figure title to clarify what comes form the ESMs (downward longwave flux, net downward shortwave flux) and what results from the RCM bulk formulae computation (net longwave, wind stress) (lines 275-281; Fig.4).

Line 316: Another example of a subjective phrase "weak dissolved O2 concentrations".

R: We modified the sentence, cited a value for the oxygen concentration, and added a reference (line 347).

Line 318: There is no entry for Espinoza et al., 2019 in the reference list.

R: We modified the reference (Espinoza-Morriberón et al., 2019).

Line 321: You mean the RCM eastward surface flow?

R: No, this is actually the ESM eastward subsurface flow, as 95°W is the location of the RCM western boundary. We modified the sentence as follows: "we first evaluate the ESM eastward subsurface flow (which enters the western boundary of the RCM) at 95°W" (line 353).

Line 328: "The trend is relatively weak. . ."

R: This sentence has been changed due to changes in the figure (see our general comment above).

Line 330: I find that the use of parentheses to indicate the opposite of an idea in a paragraph is confusing and inefficient. I invite the authors to use parentheses for clarification and citations only and not to save space. See Robock, A. 2010. Parentheses are (are not) for references and clarification (savings space). Eos, Trans. Amer. Geophys. Union, 91(45): 419).

R: The sentence has been modified due to changes in the figures (lines 360-365).

Table 1. Needs a better description of terms. What does 10 m mean? 10 m wind?

R: 10 m indicates the thickness of the ESM ocean surface layer. The legend of the table (now Table 2) has been modified .

C: Fig. 1. For clarity, I would suggest to make the vertical axis of each subplot equal and visualizing the extent of the influence of the OMZ on nitrate is not evident.

R: Fig1. Vertical axis is now 0-500 m for oxygen panel in Fig.1 Note that it is 0-250

m for temperature and salinity to better highlight differences of the thermocline and subsurface salinity maximum structures.

C: Also, the thickness of the lines representing the selected ESM's is not really different from the rest. Perhaps the legend should refer to these as "solid colored lines" instead of "thick colored lines."

R: We have increased the thickness of the lines in Figure 1.

C: Fig. 2. The description of the legend is not consistent with what is being showed and what is described on the text. i.e., b) and d) should be output from the RCM (downscaled).

R: Corrected.

Fig. 3. The word "value" is missing in the legend just before (c).

R: Corrected.

Fig. 4. In the legend (c) is missing.

R: Corrected.

Fig. 11. In the legend, fix the superscript in $\mu$mol L-1.

R: Corrected.

Fig. 16. The legend is wrong, there are no figures 16d-f.

R: Corrected.

Fig. 17. In a) the title of the figure is wrong. These should be the trends of the ESMs not RCM as mentioned in the legend and in the text (line 508). It should be indicated somewhere in the legend that the trends in b) and c) correspond to the R-GCM' sensitivity experiments.

R: Figure 17 has been modified. The results from the simulations forced by dissolved

oxygen climatological boundary conditions. Thus they correspond to the RCM' values and not to ESM values. The legend and the text have been modified accordingly.

Please also note the supplement to this comment:
https://www.biogeosciences-discuss.net/bg-2020-4/bg-2020-4-AC1-supplement.pdf

---

## Author Comment (AC2) · 5 May 2020

General Comments

The authors explore the projected physical and biogeochemical state of the Northern Humboldt Current System (NHCS) under future climate change using a regional circulation model (RCM) forced by three global earth system models (ESMs). They describe changes in a range of ocean properties from temperature to zooplankton biomass, focusing on trends relative to historical conditions as well as the differences among the different ESM and RCM projections.

Future conditions in eastern boundary upwelling systems like the NHCS are of considerable interest due to the biogeochemical, ecological, and socioeconomic importance

of these regions. It's also well known that fine scale dynamics in these regions are important and are not well captured by coarse resolution global models, so there is in terest in the potential added value provided by dynamical downscaling. Therefore, this is valuable work and is at the cutting edge of regional ocean projection. The inclusion of biogeochemistry, the use of multiple ESMs to force the regional model, and the bias correction of the forcing are all notable and positive elements of the research.

R: We thank the reviewer for his encouraging and constructive comments.

The manuscript is mostly descriptive; the authors note that further mechanistic analysis is left to future research. In my view, the most important results are the comparisons of projected changes between the global and regional models. We know global mod els have biases, but it's when the projected change is altered by downscaling that a stronger case is made for the need to downscale. The authors find that this is the case for biogeochemical, but not physical, variables. I have a number of specific comments below, but my main concerns are with several choices in the methods, detailed below.

Specific Comments

I have three main concerns on the methods:

1. (Section 2.3). The choice of which ESMs to use has been justified based on his torical comparisons with observations. However, there is a growing body of research arguing against this method, since these historical model evaluations do not neces sarily correspond to how well a model captures the response to future climate forcing. " Emergent constraints" have been offered as a more relevant method for evaluating climate models (Hall et al. 2019). In the absence compelling reasons why a model is unrealistic for the future change, the default should be to pick a suite of models that capture the range of potential futures.

R: We agree with the reviewer that the method of "emergent constraints" is a relevant method for selecting ESMs in order to project the impact of climate change on particular

variables, and that even ESMs with strong biases can be used in that method. Indeed it is possible that a model may represent a correct relation between present state and future conditions even with an important bias in the present state. If our study were to be done again today, we would probably investigate this approach and the method of emergent constraints would be a good candidate. However, in the present study we are interested in the projections of several parameters (stratification, upwelling, OMZ, productivity), thuswe would have had to find different "emergent constraints" for each of these variables, which may be intricate, and moreover, may lead to select different models for each constraint. Also, we have to admit that we were not aware of this approach at the beginning of our study, which has mainly been used in basic climate studies and not for regional downscaling (to our knowledge). Therefore, we consider that it is beyond the scope of the present work to select ESMs based on an emergent constraint which remains to be identified for the region of study, but we agree that it would be interesting to investigate further such an approach. We added a short paragraph to discuss this aspect at the beginning of the discussion (discussion section 4.1, lines 509-519).

2. (Section 2.4). As I understand it, this method produces forcing with no high- resolution (sub monthly) variability. High frequency wind variability can be very im portant especially to the BGC in Eastern Boundary systems. For example, Gruber et al (2006) attribute model chlorophyll biases to the use of monthly forcing. For future projections, one can add representative high frequency variability (e.g., from historical reanalysis) as a third term on the right hand side of equation (1). Similar has been done for historical sensitivity analyses (Frischknecht et al. 2015, Jacox et al. 2015).

R: We fully agree with the reviewer on that point: high frequency wind variability can be important in EBUS, in areas where upwelling tends to occur episodically, as stated in Gruber et al. (2006). Off central Peru, upwelling favorable winds are persistent over longer periods than for example off Central Chile or Northern California. So, we may expect a relatively moderate effect of high frequency wind variability off Peru. A

previous work by Echevin et al. (2014) showed indeed that its impact in the Peru upwelling system on some of the key biogeochemical fields is not strong. In this study, they performed sensitivity experiments on the boundary and atmospheric forcing. Two simulations were compared, one with daily wind stress (named REF, see table 2 in Echevin et al., 2014) and one with monthly climatological wind stress (named CLIM). The mean state computed over 7 years of simulation (Fig.9 in Echevin et al., 2019) displays very little change between the REF and CLIM simulations for cross- shore profiles of chlorophyll, nitrate, phosphate, silicate and iron. Thus we believe that the impact of the wind sub-monthly variability may not play an important role in this system and would not strongly impact the low frequency variability we focussed on. Note also that we had no choice but to use the monthly forcing as daily wind forcing was not available for all the ESMs we selected at the time we started our study.

The text has been modified as follows: "Note that submonthly wind variability may impact significantly surface chlorophyll in northern California (e.g. Gruber et al., 2006). However, previous regional modeling experiments in the NHCS showed a weak impact (less than 10% difference) of daily wind stress with respect to monthly wind stress on 7-year-averaged biogeochemical variables (Echevin et al., 2014). This suggests that using monthly winds may not impact significantly the climate trends reported in this study." (lines 179-184)

3. (Section 2.5): First, it's unclear why one would not bias-correct the physical ocean boundary conditions. For consistency they should be treated like the surface and ocean BGC fields. Second, oxygen should be treated the same as the other biogeochemical variables. While I understand the concern about unrealistic oxygen values, the oxygenation trend is inextricably linked to the trend in nitrate concentration (Fig. 11) and presumably other nutrients, and in turn with trends in productivity. It doesn't make sense to deem the oxygen trend unrealistic and the others realistic. Furthermore, since oxygen and nitrate variability are closely coupled, imposing the ESM change in one but not the other introduces biogeochemical inconsistencies that may compromise the

RCM findings. The analysis of oxygen using climatological boundary conditions is still interesting as it allows one to separate different contributions to the regional change, but it's not consistent with the rest of the analysis. Therefore, the main text should include the GCM change, with the context that you are trying to bound the range of possible futures, not to predict exactly what happens in the future. The oxygen analysis using climatological boundary conditions can move to discussion.

R: As suggested by Reviewer#1, we now include in Figure 1 the ESMs temperature and salinity profiles in the equatorial region. We also compute a normalized bias defined as follows: NB= (z)= |X_model(z)-X_obs(z)|/X_obs(z) x 100. This bias has been computed for each variable X=(T,S, nutrients, O2) in a new Table 1. This allows comparing the amplitude of the normalized bias between the ESMs. We also find that the normalized biases for temperature and salinity are weaker than those for nutrients and oxygen, which justified our approach. In other words, we estimate objectively that biogeochemical variables are less well represented than physical ones, which led us to not correct the bias of physical variables. Another difficulty would be to correct the bias of equatorial currents close to the equator where geostrophy is not valid. An interesting alternative could be to use a reanalysis (e.g. SODA) as a climatological present state and add ESM anomalies to this climatological state. This approach is however beyond the scope of the present study.

We also agree with the reviewer that separating oxygen from the other biogeochemical variables is not consistent as oxygen values can feedback on nitrate concentration. Thus we now present in the results section the RCM solutions obtained with the ESM oxygen trends as boundary conditions, and move results from the RCM simulations with WOA climatological oxygen conditions to the discussion section (lines 562-570). This led to many modifications in the text and figures.

Detailed Comments:

L20, 428, 547: Suggest removing "business as usual": See Hausfather and Peters

(2020).

R: We now use the term "worst case scenario".

L87: Unclear what "in the following paragraphs" refers to. The whole rest of the paper?

R: We rephrased this sentence.

L115-121: The differences are described and are stated to be important, but it's not clear what is the motivation for these changes.

R: The changes are due to the fact that the PISCES model described in Aumont et al. (2015) is a more recent version of the PISCES model (PISCES version 2). We had to use an earlier version (PISCES version 0) coupled to ROMS at the time of the study. The PISCESv2 had not been coupled to ROMS at the beginning of our study.

We modified the text as follows: " Detailed parameterizations of PISCES (version 2) are reported in Aumont et al. (2015). Note that we used an earlier version of the model (PISCESv0) in this study, as PISCESv2 had not been coupled to ROMS yet at the beginning of our study. Here we describe the following parameterizations of PISCESv0:..." (lines 119-121).

Section 2.6: The temporal coverage of these data sets is quite short for evaluating historical model performance, given that the decadal variability in the ESMs should not be expected to align with nature. Something like 30 years would be more appropriate, but in any case the authors should be wary of caveats associated with using short observational records.

R: We agree with the reviewer that a longer regional simulation over the historical time period would be appropriate to filter decadal variability. However, to avoid performing regional simulations over 150years (1950-2100), we chose to limit our historical simulations to the period 1997-2015. Following Reviewer#1's comment, we added a supplementary figure displaying the cross-shore sections of mean temperature and dissolved oxygen over 2006-2015 and a comparison with climatological observations.

After a spin-up phase of 8 years (1997-2005), the simulations are equilibrated and the biases are reasonable.

L205: This is probably fine as a proxy, but it's worth noting that it doesn't explicitly represent upwelling, including the curl-driven component. If so inclined, one could get a more accurate upwelling metric by integrating the Ekman and geostrophic components over the region of interest or by using the vertical velocity at the based of the Ekman layer (Jacox et al. 2018). It would also be helpful here to describe the calculation of the cross-shore geostrophic transport.

R: There seems to be a misunderstanding. We compute the upwelling index by integrating over the Ekman layer depth and over the region of interest the cross-shore current, which is almost exactly equal to the Ekman current and the geostrophic current (see Fig. 10 in Oerder et al., 2015). The same calculation is done by Jacox et al. (2018), who showed that the horizontal transport computed in this manner is very close to the upwelling computed using model vertical velocities. We added a reference to Jacox et al. (2018) in the manuscript (see lines 217-224).

L221: Thanks to its high spatial resolution and the bias correction of the forcing.

R: We agree. Corrected (line 240).

Fig. 3: I think it would be more appropriate to show the bias corrected ESM change (i.e., remove the mean ESM SST bias so that they all start from the same place). I also don't think a % change is best for SST, at least if the units are Celsius. In Fig. 3 the ESM % changes are lower because they are starting from a warm-biased state. But the magnitudes of projected temperature changes are as large as or larger than the RCM. Lastly, throughout the manuscript some indication of significance should be added to the trends.

R: We performed the suggested changes. Fig.3b now shows the ESMs SST evolutions starting from the same value in 2006. We agree that this shows better that the RCMs

SST follows the ESMs SST when the initial bias is corrected. We also added the SST change values (instead of percentages) in the figure. Last, we have estimated the trend uncertainty using a bootstrap method based on a 10 000 member synthetic distribution derived by randomly removing data from the annual series. The method is described in the methodology section 2.8. (lines 233-236). We converted the trend uncertainty into a percentage uncertainty, now reported in Tables 3, 4, 5. We also have computed the $R^2$ from the least square estimation in Tables 3,4,5. Most of the trends are significant at the 10% level. The significant trends are reported in bold font in the tables.

L248-249: Did Bakun actually project cooling, or just intensified upwelling? There could be intensified upwelling but still warming due to dominance of the surface heating.

R: The reviewer is right. Bakun (1990) projected intensified wind-driven upwelling and suggested that it would induce a cooler surface ocean and more foggy coastal regions. Bakun et al. (2010) only projected intensified wind-driven upwelling. We suppressed the sentence.

L252-253: It's not clear to me the evidence that this pattern results from the upwelling and subsequent lateral transport/damping of subsurface anomalies.

R: We agree with the comment. We suppressed the statement.

Fig. 4: Would be informative to see the net longwave and shortwave radiation (not just downwelling).

R: We added the net longwave radiation (now Fig. 4d). The shortwave radiation presented here is the net shortwave radiation coming from the ESM. We modified the titles of Fig.4 to clarify which forcing comes from the ESM (net shortwave and downward longwave radiation) and which results from the RCMs air-sea interactions (net longwave radiation and wind stress).

L266: Initially it seems strange that the offshore transport trend is 2x greater than the wind stress trend, since the transport is linearly related to the wind stress. But, it does

make sense because when you calculate the Ekman transport (i.e., Fig. 5a minus Fig. 5b), the change is âĹij10%, consistent with the winds. This should be explained in the text, and I suggest adding the Ekman transport as a third panel to Fig. 5.

R: We added the Ekman transport in Fig.5 (now Fig.5c). It shows clearly that the net offshore transport is equal to the sum of the Ekman transport and geostrophic transport. We modified the text accordingly (lines 287-288).

L269-270: It's also interesting that since there's a long-term trend in Ekman transport but not in geostrophic transport, the relative contribution of the geostrophic transport increases over time.

R: We agree. We added this comment in the text (line 291).

L275: Do you mean they are locally influenced by the passage of waves? Or are you suggesting the waves actually propagate (advect) the anomalies somehow?

R: We rephrased the beginning of the paragraph as follows: "Nearshore subsurface temperature anomalies are impacted by equatorial subsurface temperature anomalies in two ways: thermocline anomalies may propagate along the equatorial and coastal wave guide (e.g. Echevin et al., 2012, 2014; Espinoza-Morriberón et al., 2017, 2018), and temperature anomalies may be transported eastward and poleward by the near-equatorial subsurface jets (Fig.2a; Montes et al., 2010, 2011). The latter is particularly strong during eastern Pacific El Nino events (e.g. Colas et al. 2008 for the 1997-1998 event)." (lines 298-302).

L279: I would be careful about equating the depth of an isotherm (D20) with the depth of the thermocline (i.e., the depth of maximum temperature gradient). Temperature biases (or changes) will alter D20 but not necessarily the thermocline depth.

R: We no longer equate D20 with thermocline. We changed the title of the figures and modified the text as follows: "The thermal structure of the upper layer is strongly impacted by climate change in the eastern equatorial Pacific. The depth of the 20°C

isotherm (hereafter D20) is used to characterize the thickness of the warm surface layer. "(lines 303-304).

L297: How is MLD calculated?

R: The model surface boundary layer (hbl) is computed from the value a critical Richardson number computed using the KPP formulation. Previous modelling work show that the surface boundary layer thickness is very close to the model mixed layer (Liu and Fox-Kemper, 2017). We added this information and this reference (lines 319-321).

Figure 8: I would like to see the ESM tendencies on here as well

R: We added the ESM tendencies in Fig.8. In the figure the anomalies at 95°W are from the ESMs only as 95°W is the location of the RCM western open boundary. We modified the text accordingly (lines 333-335).

L308-309: I don't understand why this statement is here. I would delete it

R: Agreed. We suppressed the statement.

L351 and elsewhere: "deemed realistic enough" isn't very convincing. I don't think you have to argue for the realism of the ESM changes, rather you are looking at the regional impact of the ESM changes as one potential future scenario.

R: Agreed. We suppressed the statement.

L352: Since the 95W location is discussed a number of times, it would be helpful to show it on a map (e.g., Fig. 2a along with the coastal region)

R: Agreed. We added a red line in Figure 2b and added information in the legend.

L360-365: It's hard to compare a concentration in one place (Fig. 11a) with a depth level in another place (Fig. 11c), especially when arguing that one is the driver of the other. Can these be presented in a more consistent way?

R: We modified the figure and now plot the depth of the 21 $\mu$mol L -1 nitrate iso-surface at 95°W in the RCM boundary condition (Fig.11a). It is now directly comparable to Figs.11c,d. We modified the text accordingly (lines 391-393).

L389-397: I must say I'm surprised to see trends of opposite sign in the upper 10m. Surely one can't have opposite trends at different depths within the mixed layer. Per haps this is a seasonal signature, e.g., in summer the mixed layer is very shallow (âĹij5m) and increased chlorophyll in the seasonal mixed layer is driving the overall trend. But the authors should look at this in more detail to explain how chlorophyll at 2m can have an opposite trend from chlorophyll at 7m.

R: We agree with the reviewer's comment. Following his comment, we modified the figure and now plot vertical sections of the seasonal trends computed for summer and winter (Fig.13). This is quite interesting as we see clearly (1) the deeper mixed layer depth in winter than in summer, (2) a different behavior for R-CNRM due to a stronger nitrate limitation.

We added the following paragraph in the text: " The seasonal trends in R-GFDL and R-IPSL are consistent with a shoaling of the mixed layer (Fig.7), which reduces light limitation of phytoplankton growth (e.g. Echevin et al., 2008; Espinoza-Morriberón et al., 2017) and increases surface primary productivity in summer and winter. In contrast, the R- CNRM trend in the mixed layer is negative in summer. This is likely caused by the strong deepening of the nitracline in R-CNRM (Fig.11c) and the seasonality of the wind-driven upwelling. As the upward flow is weaker in summer, the upwelling of less rich waters into the mixed layer may trigger a nutrient limitation of phytoplankton growth. On the other hand, as the upward flow remains strong during winter, nutrient limitation does not occur. Light limitation of phytoplankton growth reduces because of the shoaling of the mixed layer, enhancing phytoplankton growth (as in the two other RCMs). Moreover, visual correlation between decadal variability of the chlorophyll con-tent and nitracline depthin R-CNRM (e.g. the oscillations in 2070-2100 in Fig.11c and Fig.12c) also suggests that nitrate limitation of phytoplankton growth may play a role."

(lines 334-346).

L430: This may be true, but without a heat budget it's speculative. There will be other contributions as well (e.g., local surface fluxes). The text at L456-460 is good.

R: We modified the sentence.

L547-552: There is also a summary statement like this in the previous section (L427-429). One of them should be cut – probably the earlier one.

R: We left the earlier statement as it is placed in the summary. We believe that repeating this statement in a similar manner in the conclusion section does not burden the manuscript.

L561-565: I think this is all speculation, so should be presented as hypotheses rather than fact (unless there is evidence to support it)

R: We agree. We have modified the text accordingly: " We can speculate that this happens for two reasons: the enhanced thermal stratification due to the warming may alleviate light limitation and vertical dilution, and the reduction of wind-driven offshore transport may allow plankton to accumulate near the coast. These processes could partly compensate the reduction of primary productivity due to a deeper nitracline and reduced wind-driven coastal upwelling." (lines 616-619).

Technical Corrections

L117-118: Does the period in a.T indicate multiplication?

R: Yes. We modified the text. It is now written differently (see text)

L368-369: Quasi-absent doesn't make sense. Maybe negligible? Insignificant?

R: We modified the text.

Table 1: Suggest including in the caption the meaning of abbreviations (mainly Pg and Zg) and the meaning of (10m) in the number of vertical levels column. Also I don't think

the full references are needed in the table, they can be in the reference list.

R: We modified the caption and the references in the table accordingly.

Figure 4: in caption (d) should be (c)

R: We modified the caption accordingly.

Figure 6, 7, 13, 16: Values should be positive for depth

R: We modified the figures accordingly.

Figure 17: Top panel should be ESM?

R: Fig.17 has been modified as it now shows the RCM results forced by the climatological oxygen boundary conditions.

Please also note the supplement to this comment:
https://www.biogeosciences-discuss.net/bg-2020-4/bg-2020-4-AC2-supplement.pdf